# Contributions of local speech encoding and functional connectivity to audio-visual speech perception

Bruno L Giordano[1,2]*, Robin A A Ince[2], Joachim Gross[2], Philippe G Schyns[2], Stefano Panzeri[3], Christoph Kayser[2]*

[1]Institut de Neurosciences de la Timone UMR 7289, Aix Marseille Université – Centre National de la Recherche Scientifique, Marseille, France; [2]Institute of Neuroscience and Psychology, University of Glasgow, Glasgow, United Kingdom; [3]Neural Computation Laboratory, Center for Neuroscience and Cognitive Systems, Istituto Italiano di Tecnologia, Rovereto, Italy

**Abstract** Seeing a speaker's face enhances speech intelligibility in adverse environments. We investigated the underlying network mechanisms by quantifying local speech representations and directed connectivity in MEG data obtained while human participants listened to speech of varying acoustic SNR and visual context. During high acoustic SNR speech encoding by temporally entrained brain activity was strong in temporal and inferior frontal cortex, while during low SNR strong entrainment emerged in premotor and superior frontal cortex. These changes in local encoding were accompanied by changes in directed connectivity along the ventral stream and the auditory-premotor axis. Importantly, the behavioral benefit arising from seeing the speaker's face was not predicted by changes in local encoding but rather by enhanced functional connectivity between temporal and inferior frontal cortex. Our results demonstrate a role of auditory-frontal interactions in visual speech representations and suggest that functional connectivity along the ventral pathway facilitates speech comprehension in multisensory environments.

*For correspondence: Bruno.
Giordano@glasgow.ac.uk (BLG);
christoph.kayser@glasgow.ac.uk
(CK)

**Competing interests:** The authors declare that no competing interests exist.

## Introduction

When communicating in challenging acoustic environments we profit tremendously from visual cues arising from the speakers face. Movements of the lips, tongue or the eyes convey significant information that can boost speech intelligibility and facilitate the attentive tracking of individual speakers (*Ross et al., 2007*; *Sumby and Pollack, 1954*). This multisensory benefit is strongest for continuous speech, where visual signals provide temporal markers to segment words or syllables, or provide linguistic cues (*Grant and Seitz, 1998*). Previous work has identified the synchronization of brain rhythms between interlocutors as a potential neural mechanism underlying the visual enhancement of intelligibility (*Hasson et al., 2012*; *Park et al., 2016*; *Peelle and Sommers, 2015*; *Pickering and Garrod, 2013*; *Schroeder et al., 2008*). Both acoustic and visual speech signals exhibit pseudo-rhythmic temporal structures at prosodic and syllabic rates (*Chandrasekaran et al., 2009*; *Schwartz and Savariaux, 2014*). These regular features can entrain rhythmic activity in the observer's brain and facilitate perception by aligning neural excitability with acoustic or visual speech features (*Giraud and Poeppel, 2012*; *Mesgarani and Chang, 2012*; *Park et al., 2016*; *Peelle and Davis, 2012*; *Schroeder and Lakatos, 2009*; *Schroeder et al., 2008*; *van Wassenhove, 2013*; *Zion Golumbic et al., 2013a*). While this model predicts the visual enhancement of speech encoding in challenging multisensory environments, the network organization of multisensory speech encoding remains unclear.

**eLife digest** When listening to someone in a noisy environment, such as a cocktail party, we can understand the speaker more easily if we can also see his or her face. Movements of the lips and tongue convey additional information that helps the listener's brain separate out syllables, words and sentences. However, exactly where in the brain this effect occurs and how it works remain unclear.

To find out, Giordano et al. scanned the brains of healthy volunteers as they watched clips of people speaking. The clarity of the speech varied between clips. Furthermore, in some of the clips the lip movements of the speaker corresponded to the speech in question, whereas in others the lip movements were nonsense babble. As expected, the volunteers performed better on a word recognition task when the speech was clear and when the lips movements agreed with the spoken dialogue.

Watching the video clips stimulated rhythmic activity in multiple regions of the volunteers' brains, including areas that process sound and areas that plan movements. Speech is itself rhythmic, and the volunteers' brain activity synchronized with the rhythms of the speech they were listening to. Seeing the speaker's face increased this degree of synchrony. However, it also made it easier for sound-processing regions within the listeners' brains to transfer information to one other. Notably, only the latter effect predicted improved performance on the word recognition task. This suggests that seeing a person's face makes it easier to understand his or her speech by boosting communication between brain regions, rather than through effects on individual areas.

Further work is required to determine where and how the brain encodes lip movements and speech sounds. The next challenge will be to identify where these two sets of information interact, and how the brain merges them together to generate the impression of specific words.

Previous work has implicated many brain regions in the visual enhancement of speech, including superior temporal (*Beauchamp et al., 2004*; *Nath and Beauchamp, 2011*; *Riedel et al., 2015*; *van Atteveldt et al., 2004*), premotor and inferior frontal cortices (*Arnal et al., 2009*; *Evans and Davis, 2015*; *Hasson et al., 2007b*; *Lee and Noppeney, 2011*; *Meister et al., 2007*; *Skipper et al., 2009*; *Wright et al., 2003*). Furthermore, some studies have shown that the visual facilitation of speech encoding may even commence in early auditory cortices (*Besle et al., 2008*; *Chandrasekaran et al., 2013*; *Ghazanfar et al., 2005*; *Kayser et al., 2010*; *Lakatos et al., 2009*; *Zion Golumbic et al., 2013a*). However, it remains to be understood whether visual context shapes the encoding of speech differentially within distinct regions of the auditory pathways, or whether the visual facilitation observed within auditory regions is simply fed forward to upstream areas, perhaps without further modification. Hence, it is still unclear whether the enhancement of speech-to-brain entrainment is a general mechanism that mediates visual benefits at multiple stages along the auditory pathways.

Many previous studies on this question were limited by conceptual shortcomings: first, many have focused on generic brain activations rather than directly mapping the task-relevant sensory representations (activation mapping vs. information mapping [*Kriegeskorte et al., 2006*]), and hence have not quantified multisensory influences on those neural representations shaping behavioral performance. Those who did focused largely on auditory cortical activity (*Zion Golumbic et al., 2013b*) or did not perform source analysis of the underlying brain activity (*Crosse et al., 2015*). Second, while many studies have correlated speech-induced local brain activity with behavioral performance, few studies have quantified directed connectivity along the auditory pathways to ask whether perceptual benefits are better explained by changes in local encoding or by changes in functional connectivity (but see [*Alho et al., 2014*]). And third, many studies have neglected the continuous predictive structure of speech by focusing on isolated words or syllables (but see [*Crosse et al., 2015*]). However, this structure may play a central role for mediating the visual benefits (*Bernstein et al., 2004*; *Giraud and Poeppel, 2012*; *Schroeder et al., 2008*). Importantly, given that the predictive visual context interacts with acoustic signal quality to increase perceptual benefits in adverse environments (*Callan et al., 2014*; *Ross et al., 2007*; *Schwartz et al., 2004*; *Sumby and Pollack, 1954*), one

needs to manipulate both factors to fully address this question. Fourth, most studies focused on either the encoding of acoustic speech signals in a multisensory context, or quantified brain activity induced by visual speech, but little is known about the dependencies of neural representations of the acoustic and visual components of realistic speech (but see [*Park et al., 2016*]). Overcoming these problems, we here capitalize on the statistical and conceptual power offered by naturalistic continuous speech to study the network mechanisms that underlie the visual facilitation of speech perception.

Using source localized MEG activity we systematically investigated how local representations of acoustic and visual speech signals and task-relevant directed functional connectivity along the auditory pathways change with visual context and acoustic signal quality. Specifically, we extracted neural signatures of acoustically-driven speech representations by quantifying the mutual information (MI) between the MEG signal and the acoustic speech envelope. Similarly, we extracted neural signatures of visually-driven speech representations by quantifying the MI between lip movements and the MEG signal. Furthermore, we quantified directed causal connectivity between nodes in the speech network using time-lagged mutual information between MEG source signals. Using linear modelling we then asked how each of these signatures (acoustic and visual speech encoding; connectivity) are affected by contextual information about the speakers face, by the acoustic signal to noise ratio, and by their interaction. In addition, we used measures of information theoretic redundancy to test whether the local representations of acoustic speech are directly related to the temporal dynamics of lip movements or rather reflect visual contextual information more indirectly. And finally, we asked how local speech encoding and network connectivity relate to behavioral performance.

Our results describe multiple and functionally distinct representations of acoustic and visual speech in the brain. These are differentially affected by acoustic SNR and visual context, and are not trivially explained by a simple superposition of representations of the acoustic speech and lip movement information. However, none of these local speech representations was predictive of the degree of visual enhancement of speech comprehension. Rather, this behavioral benefit was predicted only by changes in directed functional connectivity.

## Results

Participants (n = 19) were presented with continuous speech that varied in acoustic quality (signal to noise ratio, SNR) and the informativeness of the speaker's face. The visual context could be either informative (VI), showing the face producing the acoustic speech, or uninformative (VN), showing the same face producing nonsense babble (*Figure 1A,B*). We measured brain-wide activity using MEG while participants listened to eight six-minute texts and performed a delayed word recognition task. Behavioral performance was better during high SNR and an informative visual context (*Figure 2*): a repeated measures ANOVA revealed a significant effect of SNR ($F(3,54) = 36.22$, p<0.001, Huynh-Feldt corrected, $\eta^2_p = 0.67$), and of visual context ($F(1,18) = 18.95$, p<0.001, $\eta^2_p = 51$), as well as a significant interaction ($F(3,54) = 4.34$, p=0.008, $\eta^2_p = 0.19$). This interaction arose from a significant visual enhancement (VI vs VN) for SNRs of 4 and 8 dB (paired $T(18) \geq 3.00$, Bonferroni corrected p≤0.032; p>0.95 for other SNRs).

To study the neural mechanisms underlying this behavioral benefit we analyzed source-projected MEG data using information theoretic tools to quantify the fidelity of local neural representations of the acoustic speech envelope (speech MI), local representations of the visual lip movement (lip MI), as well as the directed causal connectivity between relevant regions (*Figure 1C*). For both, local encoding and connectivity, we (1) modelled the extent to which they were modulated by the experimental conditions, and we (2) asked whether they correlated with behavioral performance across conditions and with the visual benefit across SNRs (*Figure 1C*).

### Widespread speech-to-brain entrainment at multiple time scales

Speech-to-brain entrainment was quantified by the mutual information (speech MI) between the MEG time course and the acoustic speech envelope (not the speech + noise mixture) in individual frequency bands (*Gross et al., 2013*; *Kayser et al., 2015*). At the group-level we observed widespread significant speech MI in all considered bands from 0.25 to 48 Hz (FWE = 0.05), except between 18–24 Hz (*Figure 3—figure supplement 1A*). Consistent with previous results

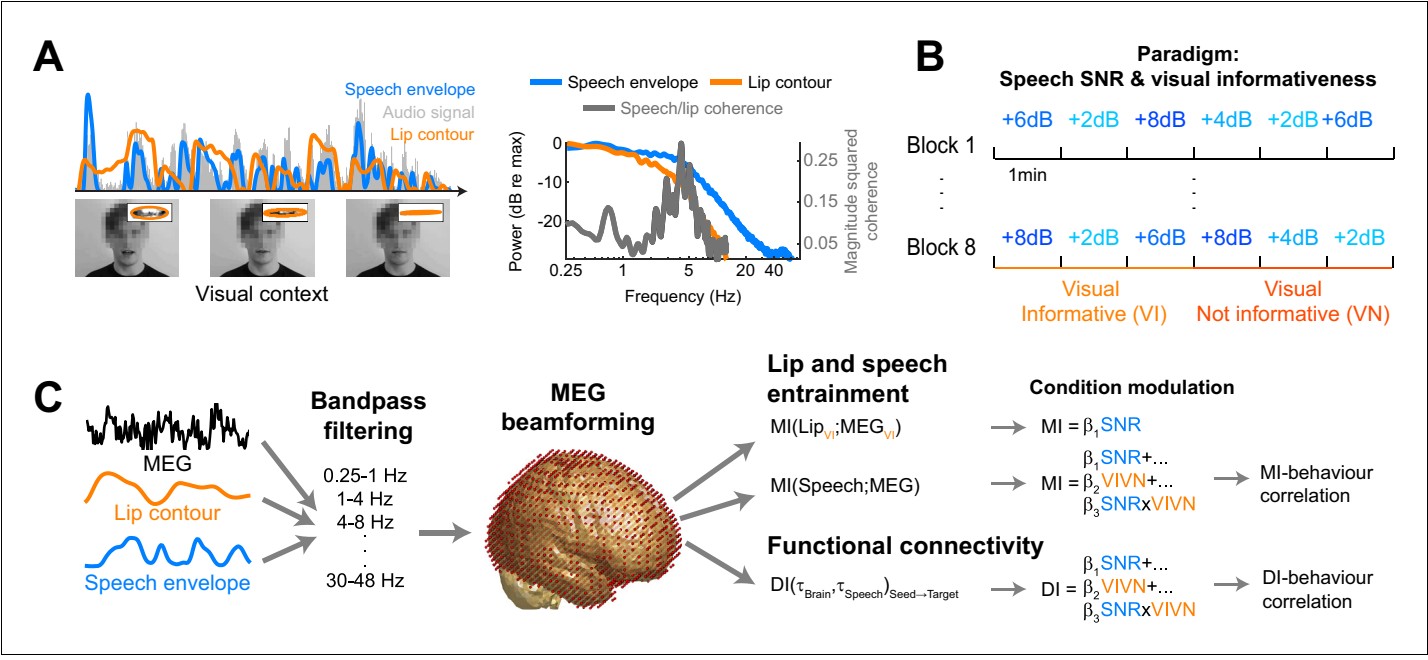

**Figure 1.** Experimental paradigm and analysis. (**A**) Stimuli consisted of 8 continuous 6 min long audio-visual speech samples. For each condition we extracted the acoustic speech envelope as well as the temporal trajectory of the lip contour (video frames, top right: magnification of lip opening and contour). (**B**) The experimental design comprised eight conditions, defined by the factorial combination of 4 levels of speech to background signal to noise ratio (SNR = 2, 4, 6, and 8 dB) and two levels of visual informativeness (VI: Visual context Informative: video showing the narrator in synch with speech; VN: Visual context Not informative: video showing the narrator producing babble speech). Experimental conditions lasted 1 (SNR) or 3 (VIVN) minutes, and were presented in pseudo-randomized order. (**C**) Analyses were carried out on band-pass filtered speech envelope and MEG signals. The MEG data were source-projected onto a grey-matter grid. One analysis quantified speech entrainment, i.e. the mutual information (MI) between the MEG data and the acoustic speech envelope (speech MI), as well as between the MEG and the lip contour (lip MI), and the extent to which these were modulated by the experimental conditions. A second analysis quantified directed functional connectivity (DI) between seeds and the extent to which this was modulated by the experimental conditions. A final analysis assessed the correlation of either MI or DI with word-recognition performance. Relevant variables in deposited data (doi:10.5061/dryad.j4567): SE_speech; LE_lip.

(*Gross et al., 2013*; *Ng et al., 2013*; *Park et al., 2016*) speech MI was higher at low frequencies and strongest below 4 Hz (*Figure 3—figure supplement 1C*). This time scale is typically associated with syllabic boundaries or prosodic stress (*Giraud and Poeppel, 2012*; *Greenberg et al., 2003*). Indeed, the average syllabic rate was 212 syllables per minute in the present material, corresponding to about 3.5 Hz. Across frequencies, significant speech MI was strongest in bilateral auditory cortex and was more extended within the right hemisphere (*Figure 3—figure supplement 1A and C*). Indeed, peak significant MI values were significantly higher in the right compared to the left hemisphere at frequencies below 12 Hz (paired t-tests; $T(18) \geq 3.1$, $p \leq 0.043$ Bonferroni corrected), and did not differ at higher frequencies ($T(18) \leq 2.78$, $p \geq 0.09$). This lateralization of speech-to-brain entrainment at frequencies below 12 Hz is consistent with previous reports (*Gross et al., 2013*).

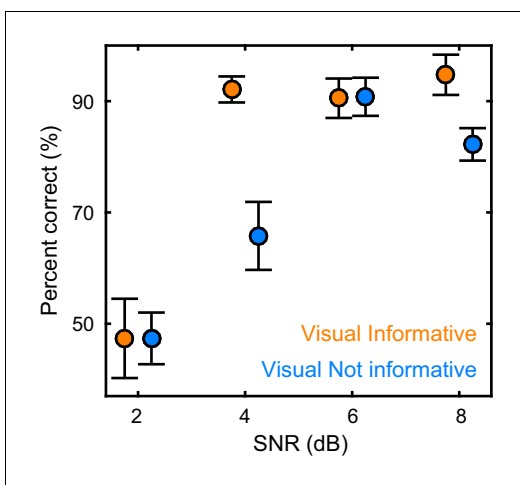

**Figure 2.** Behavioral performance. Word recognition performance for each of the experimental conditions (mean ± SEM across participants n = 19). Deposited data: BEHAV_perf.

Importantly, we observed significant speech-to-brain entrainment not only within temporal cortices but across multiple regions in the occipital, frontal and parietal lobes, consistent with the notion that speech information is represented also within motor and frontal regions (*Bornkessel-Schlesewsky et al., 2015*; *Du et al., 2014*; *Skipper et al., 2009*).

## Speech entrainment is modulated by SNR within and beyond auditory cortex

To determine the regions where acoustic signal quality and visual context affect the encoding of acoustic speech we modelled the condition-specific speech MI values based on effects of acoustic signal quality (SNR), visual informativeness (VIVN), and their interaction (SNRxVIVN). Random-effects significance was tested using a permutation procedure and cluster enhancement, correcting for multiple comparisons along all relevant dimensions. Effects of experimental factors emerged in multiple regions at frequencies below 4 Hz (*Figure 3*). Increasing the acoustic signal quality (SNR; *Figure 3A*) resulted in stronger speech MI in the right auditory cortex (1–4 Hz; local peak T statistic = 4.46 in posterior superior temporal gyrus; pSTG-R; *Table 1*), right parietal cortex (local peak T = 3.94 in supramarginal gyrus; SMG-R), and right dorso-ventral frontal cortex (IFGop-R; global peak T = 5.06). We also observed significant positive SNR effects within the right temporo-parietal and occipital cortex at 12–18 Hz (local peak right lingual gyrus, T = 5.12). However, inspection of the participant-specific data suggested that this effect was not reliable (for only 58% of participants showed a speech MI increase with SNR, as opposed to a minimum of 84% for the other SNR effects), possibly because the comparatively lower power of speech envelope fluctuations at higher frequencies (c.f. *Figure 1A*); hence this effect is not discussed further.

## Visual context reveals distinct strategies for handling speech in noise in premotor, superior and inferior frontal cortex

Contrasting informative and not-informative visual contexts revealed stronger speech MI when seeing the speakers face (VI) at frequencies below 4 Hz in both hemispheres (*Figure 3B*): the right temporo-parietal cortex (0.25–1 Hz; HG; T = 4.75; *Table 1*), bilateral occipital cortex (1–4 Hz; global T peak in right visual cortex VC-R;=6.01) and left premotor cortex (1–4 Hz; PMC-L; local T peak = 3.81). Interestingly, the condition-specific pattern of MI for VC-R was characterized by an increase in speech MI with decreasing SNR during the VI condition, pointing to a stronger visual enhancement during more adverse listening conditions. The same effect was seen in premotor cortex (PMC-L).

Since visual benefits for perception emerge mostly when acoustic signals are degraded (*Figure 2*) (*Ross et al., 2007*; *Sumby and Pollack, 1954*), the interaction of acoustic and visual factors provides a crucial test for detecting non-trivial audio-visual interactions. We found significant interactions in the 0.25–1 Hz band in the right dorso-ventral frontal lobe, which peaked in the pars triangularis (IFGt-R; T = 3.62; *Figure 3C*; *Table 1*). Importantly, investigating the SNR effect in the frontal cortex voxels revealed two distinct strategies for handling speech in noise dependent on visual context (*Figure 3D*): During VI speech MI increased with SNR in ventral frontal cortex (peak T for SNR in pars orbitalis; IFGor-R; T = 5.07), while in dorsal frontal cortex speech MI was strongest at low SNRs during VN (peak T in superior frontal gyrus; SFG-R; T = −3.55). This demonstrates distinct functional roles of ventral and dorsal prefrontal regions in speech encoding and reveals a unique role of superior frontal cortex for enhancing speech representations in a poorly informative context, such as the absence of visual information in conjunction with poor acoustic signals. For further analysis we focused on these regions and frequency bands revealed by the GLM effects (*Figure 3E*).

## Condition effects are hemisphere-dominant but not strictly lateralized

Our results reveal significantly stronger entrainment at low frequencies (c.f. *Figure 3—figure supplement 1*) and a prevalence of condition effects on speech MI in the right hemisphere (c.f. *Figure 3*). We directly tested whether these condition effects were significantly lateralized by comparing the respective GLM effects between corresponding ROIs across hemispheres (*Table 1*). This revealed that only the 1–4 Hz SNR effect in IFGop-R was significantly lateralized (T(18) = 6.03; FWE = 0.05 corrected across ROIs), while all other GLM effects did not differ significantly between hemispheres.

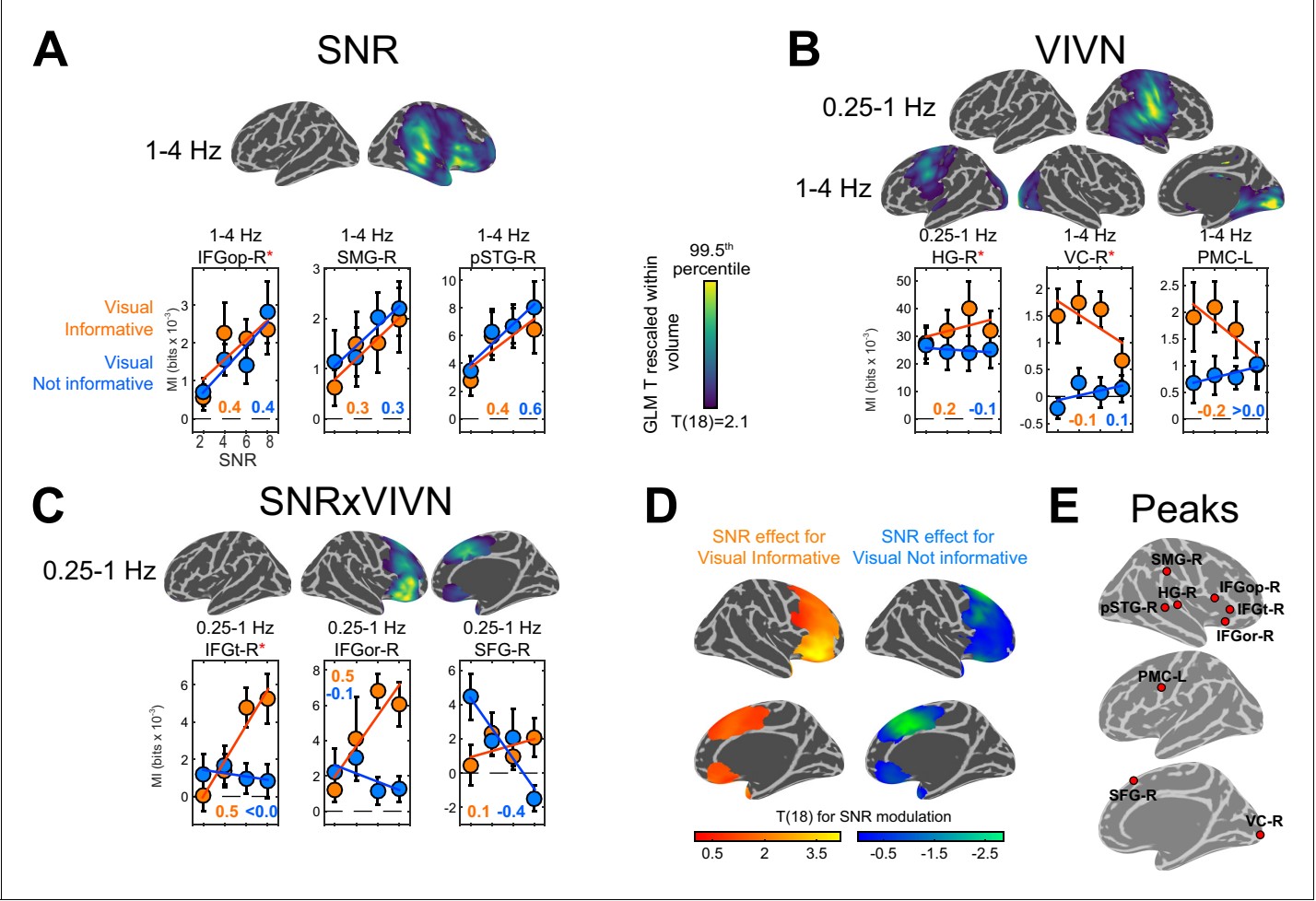

**Figure 3.** Modulation of speech-to-brain entrainment by acoustic SNR and visual informativeness. Changes in speech MI with the experimental factors were quantified using a GLM for the condition-specific speech MI based on the effects of SNR (**A**), visual informativeness VIVN (**B**), and their interaction (SNRxVIVN) (**C**). The figures display the cortical-surface projection onto the Freesurfer template (proximity = 10 mm) of the group-level significant statistics for each GLM effect (FWE = 0.05). Graphs show the average speech MI values for each condition (mean ± SEM), for local and global (red asterisk) of the T maps. Lines indicate the across-participant average regression model and numbers indicate the group-average standardized regression coefficient for SNR in the VI and VN conditions (>/ < 0.0 = positive/negative, rounded to 0). (**D**) T maps illustrating the opposite SNR effects within voxels with significant SNRxVIVN effects. MI graphs for the peaks of these maps are shown in (**C**) (IFGor-R and SFG-R = global T peaks for SNR effects in VI and VN, respectively). (**E**) Location of global and local seeds of GLM T maps, used for the analysis of directed connectivity. See also *Tables 1* and *2* and *Figure 3—figure supplements 1–2*. Deposited data: SE_meg; SE_speech; SE_miS.

The following figure supplements are available for figure 3:

**Figure supplement 1.** Entrainment of rhythmic MEG activity to the speech envelope and lip movements.

**Figure supplement 2.** Information theoretic decomposition of speech entrainment.

**Figure supplement 3.** Condition-changes in the amplitude of oscillatory activity.

## Noise invariant dynamic representations of lip movements

To complement the above analysis of speech-to-brain entrainment we also systematically analyzed the entrainment of brain activity to lip movements (lip MI). This allowed us to address whether the enhancement of the encoding of acoustic speech during an informative visual context arises from a co-representation of acoustic and visual speech information in the same regions or not. As expected based on previous work, the acoustic speech envelope and the trajectory of lip movements for the

**Table 1.** Condition effects on speech MI. The table lists global and local peaks in the GLM T-maps. Anatomical labels and Brodmann areas are based on the AAL and Talairach atlases. $\beta$ = standardized regression coefficient; SEM = standard error of the participant average. ROI-contralat. = T test for a significant difference of GLM betas between the respective ROI and its contralateral grid voxel.

| Anatomical label | Brodmann area | MNI coordinates | | | GLM effect | Frequency Band | T(18) | β(SEM) | T(18) ROI-contralat. |
|---|---|---|---|---|---|---|---|---|---|
| HG-R | 42 | 63 | −21 | 11 | VIVN | 0.25–1 Hz | 4.75* | 0.39 (0.06) | 2.00 |
| pSTG-R | 22 | 48 | −30 | 8 | SNR | 1–4 Hz | 4.46* | 0.48 (0.08) | 2.36 |
| SMG-R | 40 | 57 | −30 | 38 | SNR | 1–4 Hz | 3.94* | 0.29 (0.09) | 0.22 |
| PMC-L | 6 | −54 | 0 | 32 | VIVN | 1–4 Hz | 3.81* | 0.27 (0.06) | −0.65 |
| IFGt-R | 46 | 42 | 33 | 2 | SNRxVIVN | 0.25–1 Hz | 3.62* | 0.29 (0.07) | 1.48 |
| IFGop-R | 47 | 51 | 18 | 2 | SNR | 1–4 Hz | 5.06* | 0.36 (0.08) | 6.03* |
| IFGor-R | 47 | 30 | 26 | −16 | SNR in VI | 0.25–1 Hz | 5.07* | 0.44 (0.08) | 1.92 |
| SFG-R | 6 | 12 | 30 | 58 | SNR in VN | 0.25–1 Hz | −3.55* | −0.41 (0.09) | −2.21 |
| VC-R | 17/18 | 18 | −102 | -4 | VIVN | 1–4 Hz | 6.01* | 0.45 (0.06) | 1.84 |

*denotes significant effects (FWE = 0.05 corrected for multiple comparisons). Relevant variables in deposited data (doi:10.5061/dryad.j4567): SE_meg; SE_speech; SE_miS.

present material were temporally coherent, in particular in the delta and theta bands (*Figure 1A*) (*Chandrasekaran et al., 2009*; *Park et al., 2016*; *Schwartz and Savariaux, 2014*).

Lip-to-brain entrainment was quantified for the visual informative condition only, across the same frequency bands as considered for the speech MI (*Figure 3—figure supplement 1B*). This revealed wide-spread significant lip MI in frequency bands below 8 Hz, with the strongest lip entrainment occurring in occipital cortex (*Figure 3—figure supplement 1B*). Peak lip MI values were larger in the right hemisphere, in particular for the 4–8 Hz band (*Figure 3—figure supplement 1C*), but this effect was not significant after correction for multiple comparisons (T(18) $\leq$ 2.53, p$\geq$0.06). We then asked whether in any regions with significant lip MI the encoding of lip information changed with SNR. No significant SNR effects were found (FWE = 0.05, corrected across voxels and 0–12 Hz frequency bands), demonstrating that the encoding of lip signals is invariant across acoustic conditions. We also directly compared speech MI and lip MI within the ROIs highlighted by the condition effects on speech MI (c.f. *Figure 3E*). In most ROIs speech MI was significantly stronger than lip MI (*Table 2*; T(18) HG-R, pSTG-R, IFGop-R and PMC-L $\geq$3.58; FWE = 0.05 corrected across ROIs), while lip MI was significantly stronger in VC-R (T(18) = −3.35; FWE = 0.05).

**Table 2.** Analysis of the contribution of audio-visual signals in shaping entrainment. For each region / effect of interest (c.f. **Table 1**) the table lists the comparison of condition-averaged speech and lip MI (positive = greater speech MI); the condition effects (GLM) on the conditional mutual information (CMI) between the MEG signal and the speech envelope, while partialling out effects of lip signals; and the condition-averaged information theoretic redundancy between speech and lip MI.

| | Speech vs. lip MI | | Speech-Lip redundancy | | Speech CMI | | |
|---|---|---|---|---|---|---|---|
| Label | T(18) | Avg(SEM) | T(18) | Avg(SEM) | Effect | T(18) | β(SEM) |
| HG-R | 4.27* | 28.16 (6.59) | 0.73 | 0.33 (0.44) | VIVN | 4.37* | 0.35 (0.06) |
| pSTG-R | 3.90* | 5.42 (1.39) | 0.49 | 0.19 (0.38) | SNR | 4.66* | 0.49 (0.08) |
| SMG-R | 2.95 | 1.32 (0.45) | 1.10 | 0.51 (0.47) | SNR | 4.10* | 0.29 (0.09) |
| PMC-L | 3.58* | 1.06 (0.30) | 3.83* | 2.42 (0.63) | VIVN | 3.47* | 0.24 (0.06) |
| IFGt-R | 1.21 | 0.87 (0.72) | 2.29 | 1.75 (0.77) | SNRxVIVN | 4.07* | 0.31 (0.07) |
| IFGopR | 3.68* | 1.50 (0.41) | 4.69* | 1.56 (0.33) | SNR | 4.70* | 0.35 (0.07) |
| SFG-R | 0.88 | 0.61 (0.70) | 4.13* | 2.37 (0.57) | SNR in VN | −3.62* | −0.43 (0.09) |
| VC-R | −3.35* | −2.19 (0.65) | 2.37 | 0.68 (0.29) | VIVN | 5.77* | 0.45 (0.06) |

*denotes significant effects (FWE = 0.05 corrected for multiple comparisons). Deposited data: ID_meg; ID_speech; ID_lip; ID_infoterms.

## Speech entrainment does not reflect trivial entrainment to lip dynamics

Given that only the speech and not the lip representation were affected by SNR the above results suggest that both acoustic and visual speech signals are represented independently in rhythmically entrained brain activity. To address the interrelation between the representations of acoustic and visual speech signals more directly, we asked whether the condition effects on speech MI result from genuine changes in the encoding of the acoustic speech envelope, or whether they result from a superposition of local representations of the acoustic and the visual speech signals. Given that visual and acoustic speech are temporally coherent and offer temporally redundant information, it could be that the enhancement of speech MI during the VI condition simply results from a superposition of local representations of the visual and acoustic signals arising within the same brain region. Alternatively, it could be that the speech-to-brain entrainment reflects a representation of the acoustic speech signal that is informed by visual contextual information, but which is not a one to one reflection of the dynamics of lip movements. We performed two analyses to address this.

First, we calculated the conditional mutual information between the MEG signal and the acoustic speech envelop while partialling out the temporal dynamics common to lip movements and the speech envelope. If the condition effects on speech MI reflect changes within genuine acoustic representations, they should persist when removing direct influences of lip movements. Indeed, we found that all of the condition effects reported in *Figure 3* persisted when computed based on conditional MI (absolute T(18) ≥ 3.47; compare *Table 2* for CMI with *Table 1* for MI; ROI-specific MI and CMI values are shown in *Figure 3—figure supplement 2A,B*).

Second, we computed the information-theoretic redundancy between the local speech and lip representations. Independent representations of each speech signal would result in small redundancy values, while a common representation of lip and acoustic speech signals would reflect in a redundant representation. Across SNRs we found that these representations were significantly redundant in the ventral and dorsal frontal cortex (T(18) ≥ 3.83, for SFG-R, IFGop-R, IFGt-Rand PMC-L) but not in the temporal lobe or early auditory and visual cortices (FWE = 0.05 corrected across ROIs; *Table 2*; *Figure 3—figure supplement 2C*). However, the actual redundancy values were rather small (condition-averaged values all below 3%). All in all, this suggests that the local representations of the acoustic speech envelope in sensory regions are informed by visual evidence but in large do not represent the same information that is provided by the dynamics of lip movements. This in particular also holds for the acoustic speech MI in visual cortex. The stronger redundancy in association cortex (IFG, SFG, PMC) suggests that these regions feature co-representations of acoustic speech and lip movements.

## Directed causal connectivity within the speech network

The diversity of the patterns of speech entrainment in temporal, premotor and inferior frontal regions across conditions shown in *Figure 3* could arise from the individual encoding properties of each region, or from changes in functional connectivity between regions with conditions. To directly test this, we quantified the directed causal connectivity between these regions of interest. To this end we used Directed Information (DI), also known as Transfer Entropy, an information theoretic measure of Wiener-Granger causality (*Massey, 1990*; *Schreiber, 2000*). We took advantage of previous work that made this measure statistically robust when applied to neural data (*Besserve et al., 2015*; *Ince et al., 2017*).

We observed significant condition-averaged DI between multiple nodes of the speech network (FWE = 0.05; *Figure 4A* and *Figure 4—figure supplement 1A*). This included among others the feed-forward pathways of the ventral and dorsal auditory streams, such as from auditory cortex (HG-R) and superior temporal regions (pSTG-R) to premotor (PMC-L) and to inferior frontal regions (IFGt-R, IFGop-R), from right parietal cortex (SMG-R) to premotor cortex (PMC-L), as well as feed-back connections from premotor and inferior frontal regions to temporal regions. In addition, we also observed significant connectivity between frontal (SFG-R) and visual cortex (VC).

We then asked whether and where connectivity changed with experimental conditions (*Figure 4B*, *Table 3* and *Figure 4—figure supplement 1B*). Within the right ventral stream feed-forward connectivity from the temporal lobe (HG-R, pSTG-R) to frontal cortex (IFGt-R, IFGop-R) was enhanced during high acoustic SNR (FWE = 0.05; T(18) ≥ 3.1). More interestingly, this connectivity was further enhanced in the presence of an informative visual context (pSTG-R → IFGt-R, VIVN

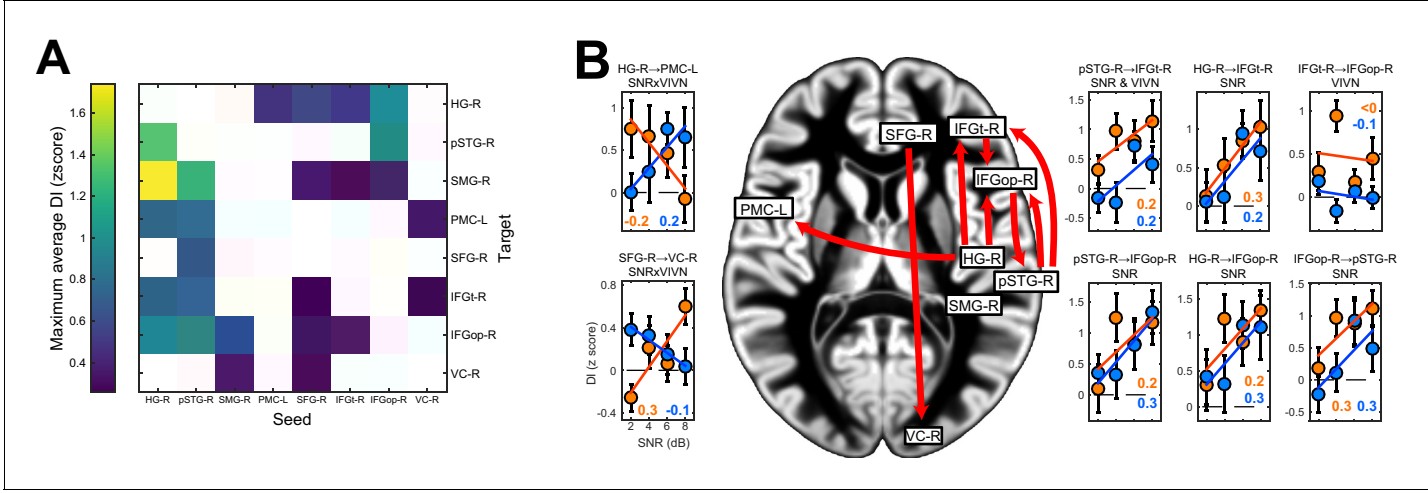

**Figure 4.** Directed causal connectivity within the speech-entrained network. Directed connectivity between seeds of interest (c.f. *Figure 3E*) was quantified using Directed Information (DI). (**A**) Maximum significant condition-average DI across lags (FWE = 0.05 across lags; white = no significant DI). (**B**) Significant condition effects (GLM for SNR, VIVN or their interaction) on DI (FWE = 0.05 across speech/brain lags and seed/target pairs). Bar graphs display condition-specific DI values for each significant GLM effect along with the across-participants average regression model (lines). Numbers indicate the group-average standardized betas for SNR in the VI and VN conditions, averaged across lags associated with a significant GLM effect (>/ < 0.0 = positive/negative, rounded to 0). Error-bars = ± SEM. See also *Table 3* and *Figure 4—figure supplement 1*. Deposited data: DI_meg; DI_speech; DI_di; DI_brainlag; DI_speechlag.

The following figure supplement is available for figure 4:

**Figure supplement 1.** Directed functional connectivity within the speech-entrained network.

effect, T = 4.57), demonstrating a direct influence of visual context on the propagation of information along the ventral stream. Interactions of acoustic and visual context on connectivity were also found from auditory (HG-R) to premotor cortex (PMC-L, negative interaction; T = −3.01). Here connectivity increased with increasing SNR in the absence of visual information and increased with decreasing SNR during an informative context, suggesting that visual information changes the qualitative nature of auditory-motor interactions. An opposite interaction was observed between the

**Table 3.** Analysis of directed connectivity (DI). The table lists connections with significant condition-averaged DI, and condition effects on DI. SEM = standard error of participant average; $\beta$ = standardized regression coefficients. T(18) = maximum T statistic within significance mask. All reported effects are significant (FWE = 0.05 corrected for multiple comparisons). Deposited data: DI_meg; DI_speech; DI_di; DI_brainlag; DI_speechlag.

| | | DI | Condition effects (GLM) | | |
|---|---|---|---|---|---|
| **Seed** | **Target** | **T(18)** | **Effect** | **T(18)** | **$\beta$(SEM)** |
| HG-R | PMC-L | 3.38 | SNRxVIVN | −3.01 | −0.24 (0.08) |
| HG-R | IFGt-R | 3.03 | SNR | 3.32 | 0.31 (0.09) |
| HG-R | IFGopR | 4.54 | SNR | 3.19 | 0.26 (0.07) |
| pSTG-R | IFGt-R | 3.39 | SNR | 3.91 | 0.32 (0.09) |
| | | | VIVN | 4.57 | 0.23 (0.05) |
| pSTG-R | IFGopR | 4.12 | SNR | 3.31 | 0.28 (0.08) |
| IFGt-R | IFGopR | 3.76 | VIVN | 3.56 | 0.21 (0.06) |
| IFGopR | pSTG-R | 4.16 | SNR | 4.65 | 0.31 (0.09) |
| SFG-R | VC-R | 4.40 | SNRxVIVN | 3.69 | 0.28 (0.08) |

frontal lobe and visual cortex (SFG-R → VC-R, T = 3.69). Finally, feed-back connectivity along the ventral pathway was significantly stronger during high SNRs (IFGt-R → pSTG-R; T = 4.56).

## Does speech entrainment or connectivity shape behavioral performance?

We performed two analyses to test whether and where changes in the local representation of speech information or directed connectivity (DI) contribute to explaining the multisensory behavioral benefits (c.f. *Figure 2*). Given the main focus on the visual enhancement of perception we implemented this analysis only for speech and not for lip MI. First, we asked where speech-MI and DI relates to performance changes across all experimental conditions (incl. changes in SNR). This revealed a significant correlation between condition-specific word-recognition performance and the strength of speech MI in pSTG-R and IFGt-R (r $\geq$ 0.28; FWE = 0.05; *Table 4* and *Figure 5A*), suggesting that stronger entrainment in the ventral stream facilitates comprehension. This hypothesis was further corroborated by a significant correlation of connectivity along the ventral stream with behavioral performance, both in feed-forward (HG-R → IFGt-R/IFGop-R; pSTG-R → IFGt-R/IFGop-R; r $\geq$ 0.24, *Table 4*) and feed-back directions (IFGop-R → pSTG-R; r = 0.37). The enhanced quality of speech perception during favorable listening conditions hence results from enhanced speech encoding and the supporting network connections along the temporal-frontal axis.

**Table 4.** Association of behavioral performance with speech entrainment and connectivity. Performance: T statistic and average of participant-specific correlation (SEM) between behavioral performance and speech MI / DI. Visual enhancement: correlation between SNR-specific behavioral benefit (VI-VN) and the respective difference in speech-MI or DI.

**Speech MI**

|        | Performance | | Visual enhancement | |
|--------|---------|----------|---------|----------|
|        | T(18)   | r(SEM)   | T(18)   | r(SEM)   |
| HG-R   | 1.27    | 0.13(0.10) | 0.21  | 0.04(0.15) |
| pSTG-R | 3.43 *  | 0.30(0.09) | 0.53  | 0.07(0.11) |
| SMG-R  | 2.35    | 0.23(0.09) | -0.39 | -0.07(0.14) |
| PMC-L  | 0.47    | 0.04(0.08) | 0.13  | 0.03(0.16) |
| IFGt-R | 3.09 *  | 0.28(0.09) | 1.25  | 0.29(0.18) |
| IFGopR | 2.38    | 0.24(0.09) | -0.25 | -0.05(0.17) |
| SFG-R  | -0.47   | -0.04(0.08) | 1.61 | 0.35(0.17) |
| VC-R   | 1.55    | 0.18(0.10) | -0.82 | -0.14(0.14) |

**Directed connectivity**

| Seed    | Target  | Performance | | Visual enhancement | |
|---------|---------|---------|----------|---------|----------|
|         |         | T(18)   | r(SEM)   | T(18)   | r(SEM)   |
| HG-R    | PMC-L   | 0.90    | 0.06(0.06) | -0.07 | -0.01(0.14) |
| HG-R    | IFGt-R  | 4.83 *  | 0.31(0.07) | 2.55 * | 0.28(0.11) |
| HG-R    | IFGopR  | 3.19 *  | 0.24(0.07) | 1.86  | 0.31(0.17) |
| pSTG-R  | IFGt-R  | 4.28 *  | 0.27(0.06) | 1.28  | 0.16(0.12) |
| pSTG-R  | IFGopR  | 3.59 *  | 0.29(0.08) | 1.82  | 0.32(0.17) |
| IFGt-R  | IFGopR  | 1.11    | 0.08(0.07) | 2.27  | 0.33(0.14) |
| IFGopR  | pSTG-R  | 4.51 *  | 0.37(0.08) | 2.55 * | 0.37(0.15) |
| SFG-R   | VC-R    | -0.04   | 0.00(0.08) | 0.90  | 0.17(0.18) |

*denotes significant effects (FWE = 0.05 corrected for multiple comparisons). Deposited data: BEHAV_perf; SE_meg; DI_meg; SE_miS; DI_di; NBC_miS; NBC_di.

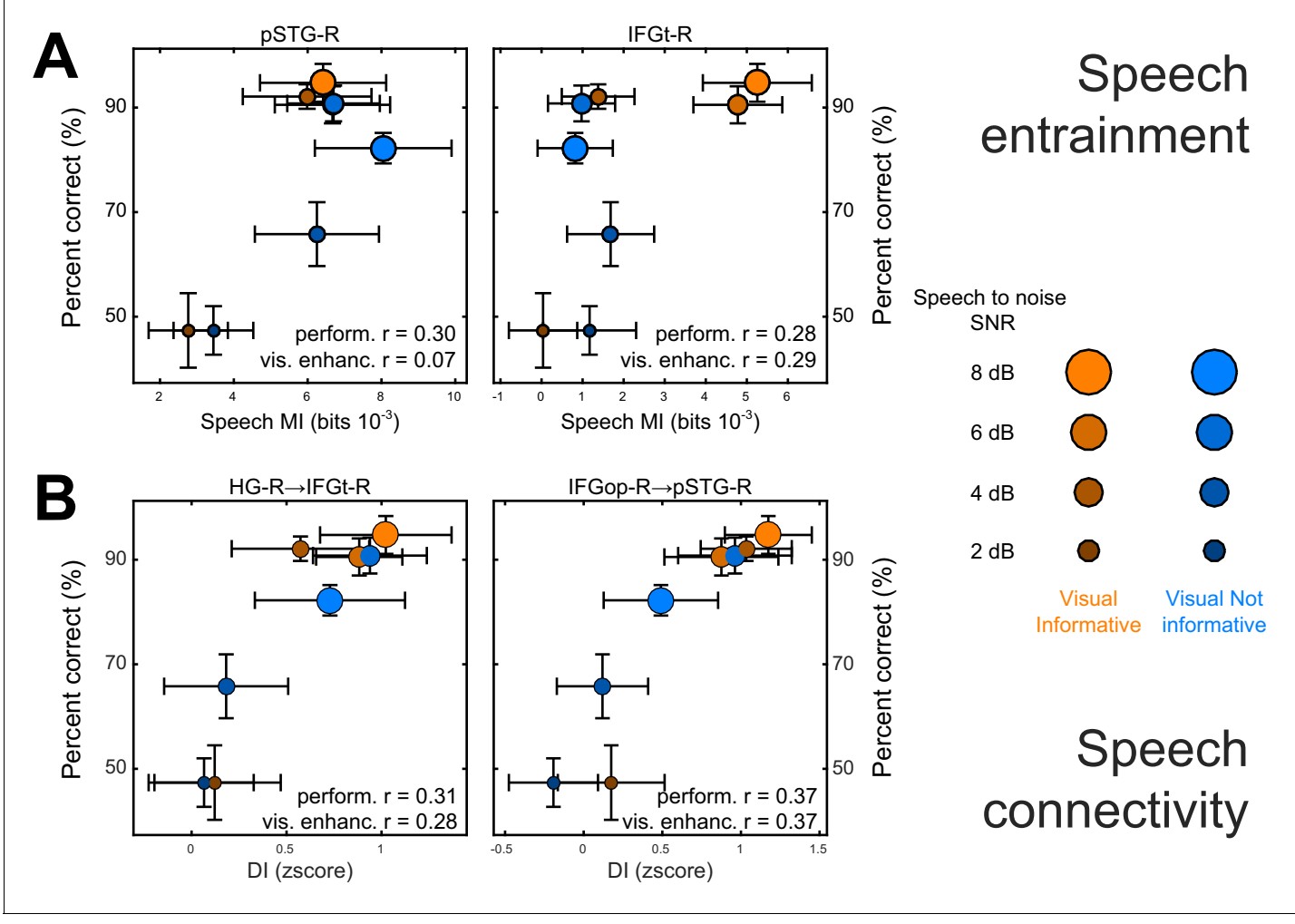

**Figure 5.** Neuro-behavioral correlations. (**A**) Correlations between behavioral performance and condition-specific speech MI (perform. (**r**), and correlations between the visual enhancement of performance and the visual enhancement in MI (vis. enhanc. (**r**). (**B**) Same for DI. Only those ROIs or connections exhibiting significant correlations are shown. error-bars = ± SEM. See also *Tables 2–3*. Deposited data: BEHAV_perf; SE_meg; DI_meg; SE_miS; DI_di; NBC_miS; NBC_di.

Second, we asked whether and where the improvement in behavioral performance with an informative visual context (VI-VN) correlates with an enhancement in speech encoding or connectivity. This revealed no significant correlations between the visual enhancement of local speech MI and perceptual benefits (all T values < FWE = 0.05 threshold; *Table 4*). However, changes in both feed-forward (HG-R → IFGt-R; r = 0.28; *Figure 5B*) and feed-back connections (IFGop-R → pSTG-R; r = 0.37) along the ventral stream were significantly correlated with the multisensory perceptual benefit (FWE = 0.05).

## Changes in speech entrainment are not a result of changes in the amplitude of brain activity

We verified that the reported condition effects on speech MI are not simply a by-product of changes in the overall oscillatory activity. To this end we calculated the condition averaged Hilbert amplitude for each ROI and performed a GLM analysis for condition effects as for speech entrainment (FWE = 0.05 with correction across ROIs and frequency bands; *Table 5*; *Figure 3—figure supplement 3*). This revealed a reduction of oscillatory activity during the visual informative condition in the occipital cortex across many bands (VC-R, 4–48 Hz), in the inferior frontal cortex (IFG-R and

**Table 5.** Changes in band-limited source signal amplitude with experimental conditions. The table lists GLM T-statistics, participant averaged standardized regression coefficients (and SEM) for significant VIVN effects on MEG source amplitude (FWE = 0.05 corrected across ROIs and frequency bands).. Effects of SNR and SNRxVIVN interactions were also tested but not significant Deposited data: SE_meg; AMP_amp.

| ROI | Band | T(18) | β(SEM) |
|---|---|---|---|
| pSTG-R | 4–8 Hz | −3.66 | −0.38 (0.09) |
| pSTG-R | 18–24 Hz | −4.11 | −0.40 (0.08) |
| IFGt-R | 24–36 Hz | −3.91 | −0.40 (0.06) |
| IFGt-R | 30–48 Hz | −4.49 | −0.39 (0.08) |
| IFGop-R | 24–36 Hz | −4.44 | −0.40 (0.07) |
| IFGop-R | 30–48 Hz | −4.14 | −0.41 (0.07) |
| VC-R | 4–8 Hz | −3.70 | −0.55 (0.08) |
| VC-R | 8–12 Hz | −4.53 | −0.70 (0.05) |
| VC-R | 12–18 Hz | −5.20 | −0.70 (0.05) |
| VC-R | 18–24 Hz | −5.57 | −0.66 (0.06) |
| VC-R | 24–36 Hz | −5.57 | −0.55 (0.08) |
| VC-R | 30–48 Hz | −4.54 | −0.46 (0.10) |

IFGop-R, 24–48 Hz), and in the pSTG-R at 4–8 Hz and 18–24 Hz. No significant effects of SNR or SNRxVIVN interactions were found (FWE = 0.05). Importantly, none of these VIVN effects overlapped with the significant changes in speech MI (0.25–4 Hz) and only the reduction in pSTG-R power overlapped with condition effects in connectivity. All in all this suggests that the reported changes in speech encoding and functional connectivity are not systematically related to changes in the strength of oscillatory activity withy acoustic SNR or visual context.

## Changes in directed connectivity do not reflect changes in phase-amplitude coupling

Cross-frequency coupling between the phase and amplitudes of different rhythmic brain signals has been implicated in mediating neural computations and communication (*Canolty and Knight, 2010*). We asked whether the above results on functional connectivity are systematically related to specific patterns of phase-amplitude coupling (PAC). We first searched for significant condition-average PAC between each pair of ROIs across a wide range of frequency combinations. This revealed significant PAC within VC-R, within pSTG-R and within SMG-R, as well as significant coupling of the 18–24 Hz VC-R power with the 0.25–1 Hz IFGop-R phase (FWE = 0.05; see *Table 6*). However, we found no

**Table 6.** Analysis of phase-amplitude coupling (PAC). The table lists the significant condition-averaged PAC values for all pairs or ROIs and frequency bands (FWE = 0.05 corrected across pairs of phase and power frequencies). SEM = standard error of participant average. None of these changed significantly with conditions (no GLM effects at FWE = 0.05). Deposited data: SE_meg.

| Phase ROI (band) | Power ROI (band) | T(18) | Pac(SEM) |
|---|---|---|---|
| pSTG-R (1–4 Hz) | pSTG-R (8–12 Hz) | 3.26 | 0.22 (0.07) |
| SMG-R (4–8 Hz) | SMG-R (30–48 Hz) | 3.58 | 0.27 (0.07) |
| IFGop-R (0.25–1 Hz) | VC-R (18–24 Hz) | 3.08 | 0.22 (0.07) |
| VC-R (4–8 Hz) | VC-R (8–12 Hz) | 3.06 | 0.35 (0.11) |
| VC-R (1–4 Hz) | VC-R (12–18 Hz) | 3.44 | 0.48 (0.13) |
| VC-R (4–8 Hz) | VC-R (24–36 Hz) | 3.76 | 0.26 (0.07) |

significant changes in PAC with experimental conditions, suggesting that the changes in functional connectivity described above are not systematically related to specific patterns of cross-frequency coupling.

## Discussion

The present study provides a comprehensive picture of how acoustic signal quality and visual context interact to shape the encoding of acoustic and visual speech information and the directed functional connectivity along speech-sensitive cortex. Our results reveal a dominance of feed-forward pathways from auditory regions to inferior frontal cortex under favorable conditions, such as during high acoustic SNR. We also demonstrate the visual enhancement of acoustic speech encoding in auditory cortex, as well as non-trivial interactions of acoustic quality and visual context in premotor and in superior and inferior frontal regions. Furthermore, our results reveal the superposition of acoustic and visual speech signals (lip movements) in association regions and the dominance of visual speech representations in visual cortex. These patterns of local encoding were accompanied by changes in directed connectivity along the ventral pathway and from auditory to premotor cortex. Yet, the behavioral benefit arising from seeing the speaker's face was not related to any region-specific visual enhancement of acoustic speech encoding. Rather, changes in directed functional connectivity along the ventral stream were predictive of the multisensory behavioral benefit.

### Entrained auditory and visual speech representations in temporal, parietal and frontal lobes

We observed functionally distinct patterns of speech-to-brain entrainment along the auditory pathways. Previous studies on speech entrainment have largely focused on the auditory cortex, where entrainment to the speech envelope is strongest (*Ding and Simon, 2013*; *Gross et al., 2013*; *Keitel et al., 2017*; *Mesgarani and Chang, 2012*; *Zion Golumbic et al., 2013a*), and only few studies have systematically compared speech entrainment along auditory pathways (*Zion Golumbic et al., 2013b*). This was in part due to the difficulty to separate distinct processes reflecting entrainment when contrasting only few experimental conditions (e.g. forward and reversed speech [*Ding and Simon, 2012*; *Gross et al., 2013*]), or based on the difficulty to separate contributions from visual (i.e. lip movements) and acoustic speech signals (*Park et al., 2016*). Based on the susceptibility to changes in acoustic signal quality and visual context, the systematic use of region-specific temporal lags between stimulus and brain response, and the systematic analysis of both acoustic and visual speech signals, we here establish entrainment as a ubiquitous mechanism reflecting distinct acoustic and visual speech representations along auditory pathways.

Entrainment to the acoustic speech envelope was reduced with decreasing acoustic SNR in temporal, parietal and ventral prefrontal cortex, directly reflecting the reduction in behavioral performance in challenging environments. In contrast, entrainment was enhanced during low SNR in superior frontal and premotor cortex. While there is strong support for a role of frontal and premotor regions in speech processing (*Du et al., 2014*; *Evans and Davis, 2015*; *Heim et al., 2008*; *Meister et al., 2007*; *Morillon et al., 2015*; *Rauschecker and Scott, 2009*; *Skipper et al., 2009*; *Wild et al., 2012*), most evidence comes from stimulus-evoked activity rather than signatures of neural speech encoding. We directly demonstrate the specific enhancement of frontal (PMC, SFG) speech representations during challenging conditions. This enhancement is not directly inherited from the temporal lobe, as temporal regions exhibited either no visual facilitation (pSTG) or visual facilitation without an interaction with SNR (HG).

We also observed significant entrainment to the temporal trajectory of lip movements in visual cortex, the temporal lobe and frontal cortex (*Figure 3—figure supplement 1*). This confirms a previous study, which has specifically focused on the temporal coherence between brain activity and lip movements (*Park et al., 2016*). Importantly, by comparing the local encoding of both the acoustic and visual speech information, and conditioning out the visual signal from the speech MI, we found that sensory cortices and the temporal lobe provide largely independent representations of the acoustic and visual speech signals. Indeed, the information theoretic redundancy between acoustic and visual representations was small and was significant only in association regions (SFG, IFG, PMC). This suggests that early sensory cortices contain largely independent representations of acoustic and visual speech information, while association regions provide a superposition of auditory and visual

speech representations. However, the condition effects on the acoustic representation in any of the analyzed regions did not disappear when factoring out the representation of lip movements, suggesting that these auditory and visual representations are differentially influenced by sensory context. These findings extend previous studies by demonstrating the co-existence of visual and auditory speech representations along auditory pathways, but also reiterate the role of PMC as one candidate region that directly links neural representations of lip movements with perception (*Park et al., 2016*).

## Multisensory enhancement of speech encoding in the frontal lobe

Visual information from the speakers' face provides multiple cues that enhance intelligibility. In support of a behavioral multisensory benefit we found stronger entrainment to the speech envelope during an informative visual context in multiple bilateral regions. First, we replicated the visual enhancement of auditory cortical representations (HG) (*Besle et al., 2008*; *Kayser et al., 2010*; *Zion Golumbic et al., 2013a*). Second, visual enhancement of an acoustic speech representation was also visible in early visual areas, as suggested by prior studies (*Nath and Beauchamp, 2011*; *Schepers et al., 2015*). Importantly, our information theoretic analysis suggests that this representation of acoustic speech is distinct from the visual representation of lip dynamics, which co-exists in the same region. The visual enhancement of acoustic speech encoding in visual cortex was strongest when SNR was low, unlike the encoding of lip movements, which was not affected by acoustic SNR. Hence this effect is most likely explained by top-down signals providing acoustic feedback to visual cortices (*Vetter et al., 2014*). Third, speech representations in ventral prefrontal cortex were selectively involved during highly reliable multisensory conditions and were reduced in the absence of the speakers face. These findings are in line with suggestions that the IFG facilitates comprehension (*Alho et al., 2014*; *Evans and Davis, 2015*; *Hasson et al., 2007b*; *Hickok and Poeppel, 2007*) and implements multisensory processes (*Callan et al., 2014*, *2003*; *Lee and Noppeney, 2011*), possibly by providing amodal phonological, syntactic and semantic processes (*Clos et al., 2014*; *Ferstl et al., 2008*; *McGettigan et al., 2012*). Previous studies often reported enhanced IFG response amplitudes under challenging conditions (*Guediche et al., 2014*). In contrast, by quantifying the fidelity of speech representations, we here show that speech encoding is generally better during favorable SNRs. This discrepancy is not necessarily surprising, if one assumes that IFG representations are derived from those in the temporal lobe, which are also more reliable during high SNRs. Noteworthy, however, we found that speech representations within ventral IFG are selectively stronger during an informative visual context, even when discounting direct co-representations of lip movements. We thereby directly confirm the hypothesis that IFG speech encoding is enhanced by visual context.

Furthermore, we demonstrate the visual enhancement of speech representations in premotor regions, which could implement the mapping of audio-visual speech features onto articulatory representations (*Meister et al., 2007*; *Morillon et al., 2015*; *Morís Fernández et al., 2015*; *Skipper et al., 2009*; *Wilson et al., 2004*). We show that that this enhancement is inversely related to acoustic signal quality. While this observation is in agreement with the notion that perceptual benefits are strongest under adverse conditions (*Ross et al., 2007*; *Sumby and Pollack, 1954*), there was no significant correlation between the visual enhancement of premotor encoding and behavioral performance. Our results thereby deviate from previous work that has suggested a driving role of premotor regions in shaping intelligibility (*Alho et al., 2014*; *Osnes et al., 2011*). Rather, we support a modulatory influence of auditory-motor interactions (*Alho et al., 2014*; *Callan et al., 2004*; *Hickok and Poeppel, 2007*; *Krieger-Redwood et al., 2013*; *Morillon et al., 2015*). In another study we recently quantified dynamic representations of lip movements, calculated when discounting influences of the acoustic speech, and reported that left premotor activity was significantly predictive of behavioral performance (*Park et al., 2016*). One explanation for this discrepancy may be the presence of a memory component in the present behavioral task, which may engage other brain regions (e.g. IFG) more than other tasks. Another explanation could be that premotor regions contain, besides an acoustic speech representation described here, complementary information about visual speech that is not directly available in the acoustic speech contour, and is either genuinely visual or correlated with more complex acoustic properties of speech. Further work is required to disentangle the multisensory nature of speech encoding in premotor cortex.

Finally, our results highlight an interesting role of the superior frontal gyrus, where entrainment was strongest when sensory information was most impoverished (low SNR, visual not informative) or when the speakers face was combined with clear speech (high SNR, visual informative). Superior frontal cortex has been implied in high level inference processes underlying comprehension, sentence level integration or the exchange with memory (*Ferstl et al., 2008*; *Hasson et al., 2007a*; *Yarkoni et al., 2008*) and is sometimes considered part of the broader semantic network (*Binder et al., 2009*; *Gow and Olson, 2016*; *Price, 2012*). Our data show that the SFG plays a critical role for speech encoding under challenging conditions, possibly by mediating sentence-level processes during low SNRs or the comparison of visual prosody with acoustic inputs in multisensory contexts.

## Multisensory behavioral benefits arise from distributed network mechanisms

To understand whether the condition-specific patterns of local speech representations emerge within each region, or whether they are possibly established by network interactions, we investigated the directed functional connectivity between regions of interest. While many studies have assessed the connectivity between auditory regions (e.g. [*Abrams et al., 2013*; *Chu et al., 2013*; *Fonteneau et al., 2015*; *Park et al., 2015*]), few have quantified the behavioral relevance of these connections (*Alho et al., 2014*).

We observed significant intra-hemispheric connectivity between right temporal, parietal and frontal regions, in line with the transmission of speech information from the temporal lobe along the auditory pathways (*Bornkessel-Schlesewsky et al., 2015*; *Hickok, 2012*; *Poeppel, 2014*). Supporting the idea that acoustic representations are progressively transformed along these pathways we found that the condition-specific patterns of functional connectivity differed systematically along the ventral and dorsal streams. While connectivity along the ventral stream was predictive of behavioral performance and strongest during favorable listening conditions, the inter-hemispheric connectivity to left premotor cortex was strongest during adverse multisensory conditions, i.e. when seeing the speakers face at low SNR. Interestingly, this pattern of functional connectivity matches the pattern of speech entrainment in PMC, reiterating the selective and distinctive contribution of premotor regions in speech encoding during multisensory conditions (*Park et al., 2016*). Our results therefore suggest that premotor representations are informed by auditory regions (HG, pSTG), rather than being driven by the frontal lobe, an interpretation that is supported by previous work (*Alho et al., 2014*; *Gow and Olson, 2016*; *Osnes et al., 2011*).

We also observed a non-trivial pattern of connectivity between the SFG and visual cortex. Here the condition-specific pattern of connectivity was similar to the pattern of entrainment in the SFG, suggesting that high-level inference processes or sentence-level integration of information in the SFG contribute to the feed-back transmission of predictive information to visual cortex (*Vetter et al., 2014*). For example, the increase of connectivity with decreasing SNR during the visual non-informative condition could serve to minimize the influence of visual speech information when this is in apparent conflict with the acoustic information in challenging environments (*Morís Fernández et al., 2015*).

Across conditions behavioral performance was supported both by an enhancement of speech representations along the ventral pathway as well as enhanced functional connectivity. This enhanced functional connectivity emerged both along feed-forward and feed-back directions between temporal and inferior frontal regions, and was strongest (in effect size) along the feed-back route. This underlines the hypothesis that recurrent processing, rather than a simple feed-forward sweep, is central to speech intelligibility (*Bornkessel-Schlesewsky et al., 2015*; *Hickok, 2012*; *Poeppel, 2014*). Central to the scope of the present study, however, we found that no single region-specific effect could explain the visual behavioral benefit. Rather, the benefit arising from seeing the speakers face was significantly correlated with the enhancement of recurrent functional connectivity along the ventral stream (HG → IFG → pSTG). Our results hence point to a distributed origin of the visual enhancement of speech intelligibility. As previously proposed (*Besle et al., 2008*; *Ghazanfar et al., 2005*; *Ghazanfar and Schroeder, 2006*; *Kayser et al., 2010*; *Zion Golumbic et al., 2013a*) this visual enhancement involves early auditory regions, but as we show here, also relies on the recurrent transformation of speech representations between temporal and frontal regions.

## A lack of evidence for lateralized representations

While the effects of experimental conditions on speech MI dominated in the right hemisphere we found little evidence that these effects were indeed significantly stronger in one hemisphere. Indeed, only the SNR effect in IFGop was significantly lateralized, while all other effects were comparable between hemispheres. Hence care needs to be taken when interpreting our results as evidence for a lateralization of speech encoding. At the same time we note that a potential right dominance of speech entrainment is in agreement with the hypothesis that right temporal regions extract acoustic information predominantly on the syllabic and prosodic time scales (*Giraud and Poeppel, 2012*; *Poeppel, 2003*). Further, several studies have shown that the right hemisphere becomes particularly involved in the representation of connected speech (*Alexandrou et al., 2017*; *Bourguignon et al., 2013*; *Fonteneau et al., 2015*; *Horowitz-Kraus et al., 2015*), and one previous study directly demonstrated the prevalence of speech-to-brain entrainment in delta and theta bands in the right hemisphere during continuous listening (*Gross et al., 2013*). This makes it little surprising that the right hemisphere becomes strongly involved in representing continuous multisensory speech. Furthermore, we a bias towards the right hemisphere may in part also be a by-product of the use of entrainment as a n index to characterize speech encoding, given that the signal power of acoustic and visual speech is highest at low frequencies (c.f. *Figure 1*), and given that the right hemisphere supposedly has a preference for speech information at long time scales (*Giraud and Poeppel, 2012*; *Poeppel, 2003*).

## The mechanistic underpinnings of audio-visual speech encoding

Speech perception relies on mechanisms related to predictive coding, in order to fill in acoustically masked signals and to exploit temporal regularities and cross-modal redundancies to predict when to expect what type of syllable or phoneme (*Chandrasekaran et al., 2009*; *Peelle and Sommers, 2015*; *Tavano and Scharinger, 2015*). Predictions modulate auditory evoked responses in an area specific manner, involve both the ventral and dorsal pathways (*Kandylaki et al., 2016*; *Sohoglu and Chait, 2016*), and affect both feedforward and feedback connections (*Auksztulewicz and Friston, 2016*; *Chennu et al., 2016*). While an informative visual context facilitates the correction of predictions about expected speech using incoming multisensory evidence, we can only speculate about a direct link between the reported effects and predictive processes. Previous studies have implied delta band activity and the dorsal auditory stream in mediating temporal predictions (*Arnal and Giraud, 2012*; *Arnal et al., 2011*; *Kandylaki et al., 2016*). Hence, the changes in delta speech entrainment across conditions seen here may well reflect changes related to the prevision of temporal predictions.

Several computational candidate mechanisms have been proposed for how multisensory information could be integrated at the level of neural populations (*Ohshiro et al., 2011*; *Pouget et al., 2002*; *van Atteveldt et al., 2014*). The focus on rhythmic activity in the present study lends itself to suggest a key role of the phase resetting of oscillatory process, as proposed previously (*Schroeder et al., 2008*; *Thorne and Debener, 2014*; *van Atteveldt et al., 2014*). However, given the indirect nature of the neuroimaging signals the present study can't rule in or out the involvement of specific neural processes.

## Conclusion

Our results provide a network view on the dynamic speech representations in multisensory environments. While premotor and superior frontal regions are specifically engaged in the most challenging environments, the visual enhancement of comprehension at intermediate SNRs seems to be mediated by interactions within the core speech network along the ventral pathway. Such a distributed neural origin of multisensory benefits is in line with the notion of a hierarchical organization of multisensory processing, and the idea that comprehension is shaped by network connectivity more than the engagement of particular brain regions.

# Materials and methods

Nineteen right handed healthy adults (10 females; age from 18 to 37) participated in this study. Subject sample size was based on previous MEG/EEG studies that contrasted speech MI derived from

rhythmic brain activity between conditions (19 and 22 participants in [*Gross et al., 2013*; *Park et al., 2016*], respectively). All participants were tested for normal hearing, were briefed about the nature and goal of this study, and received financial compensation for their participation. The study was conducted in accordance with the Declaration of Helsinki and was approved by the local ethics committee (College of Science and Engineering, University of Glasgow). Written informed consent was obtained from all participants.

## Stimulus material

The stimulus material consisted of audio-visual recordings based on text transcripts taken from publicly available TED talks also used in a previous study (*Kayser et al., 2015*) (*Figure 1A*; see also [*Park et al., 2016*]). Acoustic (44.1 kHz sampling rate) and video recordings (25 Hz frame rate, 1920 by 1080 pixels) were obtained while a trained male native English speaker narrated these texts. The root mean square (RMS) intensity of each audio recording was normalized using 6 s sliding windows to ensure a constant average intensity. Across the eight texts the average speech rate was 160 words (range 138–177) per minute, and the syllabic rate was 212 syllables (range 192–226) per minute.

## Experimental design and stimulus presentation

We presented each of the eight texts as continuous 6 min sample, while manipulating the acoustic quality and the visual relevance in a block design within each text (*Figure 1B*). The visual relevance was manipulated by either presenting the video matching the respective speech (visual informative, VI) or presenting a 3 s babble sequence that was repeated continuously (visual not informative, VN), and which started and ended with the mouth closed to avoid transients. The signal to noise ratio (SNR) of the acoustic speech was manipulated by presenting the speech on background cacophony of natural sounds and scaling the relative intensity of the speech while keeping the intensity of the background fixed. We used relative SNR values of +8, +6, +4 and +2 dB RMS intensity levels. The acoustic background consisted of a cacophony of naturalistic sounds, created by randomly superimposing various naturalistic sounds from a larger database (using about 40 sounds at each moment in time) (*Kayser et al., 2016*). This resulted in a total of 8 conditions (four SNR levels; visual informative or irrelevant) that were introduced in a block design (*Figure 1B*). The SNR changed from minute to minute in a pseudo-random manner (12 one minute blocks per SNR level). Visual relevance was manipulated within 3 min sub-blocks. Texts were presented with self-paced pauses. The stimulus presentation was controlled using the Psychophysics toolbox in Matlab (*Brainard, 1997*). Acoustic stimuli were presented using an Etymotic ER-30 tubephone (tube length = 4 m) at 44.1 kHz sampling rate and an average intensity of 65 dB RMS level, calibrated separately for each ear. Visual stimuli were presented in grey-scale and projected onto a translucent screen at 1280 × 720 pixels at 25 fps covering a field of view of 25 × 19 degrees.

Subjects performed a delayed comprehension tasks after each block, whereby they had to indicate whether a specific word (noun) was mentioned in the previous text (six words per text) or not (six words per text) in a two alternative forced choice task. The words chosen from the presented text were randomly selected and covered all eight conditions. The average performance across all trials was 73 ± 2% correct (mean and SEM across subjects), showing that subjects indeed paid attention to the stimulus. Behavioral performance for the words contained in the presented text was averaged within each condition, and analyzed using a repeated measures ANOVA, with SNR and VIVN as within-subject factors. By experimental design, the false alarm rate, i.e. the number of mistaken recognitions of words that were not part of the stimulus, was constant across experimental conditions. As a consequence, condition-specific d' measures of word recall were strongly correlated with condition-specific word-recall performance (mean correlation and SEM across subjects = 0.97 ± 0.06; T(18) for significant group-average Fisher-Z transformed correlation = 32.57, p<0.001).

## Pre-processing of speech envelope and lip movements

We extracted the envelope of the speech signal (not the speech plus background mixture) by computing the wide-band envelope at 150 Hz temporal resolution as in previous work (*Chandrasekaran et al., 2009*; *Kayser et al., 2015*). The speech signal was filtered (fourth order Butterworth filter; forward and reverse) into six frequency bands (100 Hz - 4 kHz) spaced to cover

equal widths on the cochlear map. The wide-band envelope was defined as the average of the Hilbert envelopes of these band-limited signals (*Figure 1A*). The temporal trajectory of the lip contour was extracted by first identifying the lips based on their hue and then detecting the area of mouth-opening between the lips (*Park et al., 2016*). For each video frame, the mouth aperture was subsequently estimated as the area covered by an ellipsoid fit to the detected lip contours, which was then resampled to 150 Hz for further analysis (*Figure 1A*). We estimated the coherence between the speech envelope and lip contour using spectral analysis (*Figure 1A*).

## MEG data collection

MEG recordings were acquired with a 248-magnetometers whole-head MEG system (MAGNES 3600 WH, 4-D Neuroimaging) at a sampling rate of 1017.25 Hz. Participants were seated upright. The position of five coils, marking fiducial landmarks on the head of the participants, was acquired at the beginning and at the end of each block. Across blocks, and participants, the maximum change in their position was 3.6 mm, on average (STD = 1.2 mm).

## MEG pre-processing

Analyses were carried out in Matlab using the Fieldtrip toolbox (*Oostenveld et al., 2011*), SPM12, and code for the computation of information-theoretic measures (*Ince et al., 2017*). Block-specific data were pre-processed separately. Infrequent SQUID jumps (observed in 1.5% of the channels, on average) were repaired using piecewise cubic polynomial interpolation. Environmental magnetic noise was removed using regression based on principal components of reference channels. Both the MEG and reference data were filtered using a forward-reverse 70 Hz FIR low-pass (−40 dB at 72.5 Hz); a 0.2 Hz elliptic high-pass (−40 dB at 0.1 Hz); and a 50 Hz FIR notch filter (−40 dB at 50 ± 1 Hz). Across participants and blocks, 7 MEG channels were discarded as they exhibited a frequency spectrum deviating consistently from the median spectrum (shared variance <25%). For analysis signals were resampled to 150 Hz and once more high-pass filtered at 0.2 Hz (forward-reverse elliptic filter). ECG and EOG artefacts were subsequently removed using ICA in fieldtrip (runica, 40 principal components), and were identified based on the time course and topography of IC components (*Hipp and Siegel, 2013*).

## Structural data and source localization

High resolution anatomical MRI scans were acquired for each participant (voxel size = 1 mm$^3$) and co-registered to the MEG data using a semi-automated procedure. Anatomicals were segmented into grey and white matter and cerebro-spinal fluid (*Ashburner and Friston, 2005*). The parameters for the affine registration of the anatomical to the MNI template were estimated, and used to normalize the grey matter probability maps of each individual to the MNI space. A group MNI source-projection grid with a resolution of 3 mm was prepared including only voxels associated with a group-average grey-matter probability of at least 0.25. The projection grid excluded various subcortical structures, identified using the AAL atlas (e.g., vermis, caudate, putamen and the cerebellum). Leadfields were computed based on a single shell conductor model. Time-domain projections were obtained on a block-by-block basis using LCMV spatial filters (regularization = 5%). A different LCMV filter was used for each frequency band by computing the sensor covariance for the band-pass filtered sensor signals. Further analyses focused on the maximum-variance orientation of each dipole.

## Analysis of speech and lip to brain entrainment

Motivated by previous work (*Gross et al., 2013*; *Ng et al., 2013*), we considered eight partly overlapping frequency bands (0.25–1 Hz, 1–4 Hz, 4–8 Hz, 8–12 Hz, 12–18 Hz, 18–24 Hz, 24–36 Hz, and 30–48 Hz), and isolated these from the full-spectrum MEG signals, the speech envelope and the lip trajectory in each band using a forward-reverse fourth order Butterworth filter (magnitude of frequency response at band limits = −6 dB). Entrainment was quantified using the mutual information (MI) between the filtered MEG and speech envelope or lip time courses:

$$MI\_speech = MI(MEG; speech) \quad and \quad MI\_lip = MI(MEG; lip) \tag{1}$$

The MI was calculated using a bin-less approach based on statistical copulas, which provides greater sensitivity than methods based on binned signals (*Ince et al., 2017*).

To quantify the entrainment of brain activity to the speech envelope / lip movement we first determined the optimal time lag between MEG signals and the stimulus for individual bands and source voxels using a permutation-based RFX estimate. Lag estimates were obtained based on a quadratic fit, excluding lags with insignificant MI (permutation-based FDR = 0.01). Voxels without an estimate were assigned the median estimate within the same frequency band, and volumetric maps of the optimal lags were smoothed with a Gaussian (FWHM = 10 mm). Speech / lip MI were then estimated for each band and voxel using the optimal lag. The significance of group-level MI values was assessed within a permutation-based RFX framework that relied on MI values corrected for bias at the single-subject level, and on cluster mass enhancement of the test statistics corrected for multiple comparisons at the second level (*Maris and Oostenveld, 2007*). At the single-subject level, null distributions were obtained by shuffling the assignment of stimulus and MEG, independently for each participant, that is, by permuting the six speech segments within each of the eight experimental conditions (using the same permutation across bands). Participant-specific bias-corrected MI values were then defined as the actual MI minus the median MI across all 720 possible null permutations. Group-level RFX testing relied on T-statistics for the null-hypothesis that the participant-averaged bias-corrected MI was significantly larger than zero. To this end we generated 10,000 samples of the group-averaged MI from the participant-specific null distributions, used cluster-mass enhancement across voxels and frequencies (cluster-forming threshold T(18) = 2.1) to extract the maximum cluster T across frequency bands and voxels, and considered as significant a cluster-enhanced T statistic higher than the 95th percentile of the permutation distribution (corresponding to FWE = 0.05). Significant speech MI was determined across all conditions, whereas significant lip MI was derived only for the VI condition.

To determine whether and where speech / lip entrainment was modulated by the experimental factors we used a permutation-based RFX GLM framework (*Winkler et al., 2014*). For each participant individually we considered the condition-specific bias-corrected MI averaged across repetitions and estimated the coefficients of a GLM for predicting MI based on SNR (2, 4, 6, 8 dB), VIVN (1 = Visual Informative; −1 = Visual Not informative), and their interaction; for lip MI we only considered the SNR effect in the VI condition. We computed a group-level T-statistic for assessing the hypothesis that the across-participant average GLM coefficient was significantly different than zero, using cluster-mass enhancement across voxels and frequencies. Permutation testing relied on the Freedman-Lane procedure (*Freedman and Lane, 1983*). Independently for each participant and GLM effect, we estimated the parameters of a reduced GLM that includes all of the effects but the one to be tested and extracted the residuals of the prediction. We then permuted the condition-specific residuals and extracted the GLM coefficient for the effect of interest estimated for these reshuffled residuals. We obtained a permutation T statistic for the group-average GLM coefficient of interest using the max-statistics. We considered as significant T values whose absolute value was higher than the 95th percentile of the absolute value of 10,000 permutation samples, correcting for multiple comparisons across voxels / bands (FWE = 0.05). We only considered significant GLM effects in conjunction with a significant condition-average entrainment.

## Analysis of directed functional connectivity

To quantify directed functional connectivity we relied on the concept of Wiener-Granger causality and its information theoretic implementation known as Transfer Entropy or directed information (DI) (*Massey, 1990*; *Schreiber, 2000*; *Vicente et al., 2011*; *Wibral et al., 2011*). Directed information in its original formulation (*Massey, 1990*) (termed DI[*] here) quantifies causal connectivity by measuring the degree to which the past of a seed predicts the future of a target signal, conditional on the past of the target, defined at a specific lag ($\tau_{Brain}$):

$$DI^{*}(\tau_{Brain}) = I(Target_t; Seed_{t-\tau} | Target_{t-\tau}) \tag{2}$$

While DI[*] provides a measure of the overall directed influence from seed to target, it can be susceptible to statistical biases arising from limited sampling, common inputs or signal auto-correlations (*Besserve et al., 2015*, *2010*; *Ince et al., 2017*; *Panzeri et al., 2007*). We regularized and made this measure more conservative by subtracting out values of DI computed at fixed values of speech

envelope. This subtraction removes terms – including the statistical biases described above – that cannot possibly carry speech information (because they are computed at fixed speech envelope). This results in an estimate that is more robust and more directly related to changes in the sensory input than classical transfer entropy (the same measure was termed directed feature information in [*Ince et al., 2017*, *Ince et al., 2015*]). DI was defined here as

$$DI\left(\tau_{Brain}, \tau_{Speech}\right) = DI^*\left(\tau_{Brain}\right) - DI^*\left(\tau_{Brain}\right)|Speech(\tau_{Speech}) \quad (3)$$

where DI*|Speech denotes the DI* conditioned on the speech envelope. Positive values of DI indicate directed functional connectivity between seed and target at a specific brain ($\tau_{Brain}$) and speech lag ($\tau_{Speech}$). The actual DI values were furthermore Z-scored against random effects for added robustness, which facilitates statistical comparisons between conditions across subjects (*Besserve et al., 2015*). To this end DI, as estimated for each participant and connection from *Equation 3*, was Z-scored against the distribution of DI values obtained from condition-shuffled estimates (using the same randomization procedure as for MI). DI was computed for speech lags between 0 and 500 ms and brain lags between 0 and 250 ms, at steps of one sample (1/150 Hz). We estimated DI on the frequency range of 0.25–8 Hz (forward-reverse fourth order Butterworth filter), which spans all the frequencies relevant for the condition effects on speech MI (*Figure 3*). The use of a single frequency band for the connectivity analysis greatly reduced the computational burden and statistical testing compared to the use of multiple bands, while the use of a larger bandwidth here also allowed for greater robustness of underlying estimators (*Besserve et al., 2010*). Furthermore, we computed DI by considering the bivariate MEG response defined by the band-passed source signal and its first-order difference, as this offers additional statistical robustness (*Ince et al., 2017*, *2016*). Seeds for the DI analysis were the global and local peaks of the GLM-T maps quantifying the SNR, VIVN and SNRxVIVN modulation of entrainment, and the SFG-R voxel characterized by the peak negative effect of SNR in the visual informative condition, for a total of 8 seeds (*Table 1* and *Figure 3E*). To test for the significance of condition-average DI we used the same permutation-based RFX approach as for speech MI, testing the hypothesis that bias-corrected DI > 0. We used 2D cluster-mass enhancement of the T statistics within speech/brain lag dimensions correcting for multiple comparisons across speech and brain lags (FWE = 0.05). To test for significant DI effects with experimental conditions we relied on the same GLM strategy as for MI effects, again with the same differences pertaining to cluster enhancement and comparison correction (FWE = 0.05 across lags and seed/target pairs). We only considered DI modulations in conjunction with a significant condition-average DI.

## Neuro-behavioral correlations

We used a permutation-based RFX approach to assess (1) whether an increase in condition-specific speech-MI or DI was associated with an increase in behavioral performance, and (2) whether the visual enhancement (VI-VN) of speech MI or DI was associated with stronger behavioral gains. We focused on the eight regions used as seeds for the DI analysis (c.f. *Figure 3E*). For speech MI we initially tested whether the participant-average Fisher Z-transformed correlation between condition-specific performance and speech-MI was significantly larger than zero. Uncorrected p-values were computed using the percentile method, where FWE = 0.05 p-values corrected across regions were computed using maximum statistics. We subsequently tested the positive correlation between SNR-specific visual gains (VI-VN) in speech-MI and behavioral performance using the same approach, but considered only those regions characterized by a significant condition-specific MI/performance association. For DI, we focused on those lags characterized by a significant SNR, VIVN, or SNRxVIVN DI modulation. Significance testing proceeded as for speech MI, except that Z-transformed correlations were computed independently for each lag and then averaged across lags (FWE = 0.05 corrected across all seed/target pairs).

## Analysis of the lateralization of entrainment and entrainment modulation effects

We tested for a significant lateralization of the GLM effects on speech MI reported in *Figure 3*. To this end we extracted participant specific GLM betas for each effect in the respective ROI and band. We then extracted the same GLM coefficient for the contralateral voxel and computed the between-hemispheric difference. This was tested for significance using a two-sided RFX test based on a sign-

permutation of the across-participant T value (10,000 permutations), with maximum-statistic multiple comparison correction across ROIs (FWE = 0.05; *Table 1*).

## Decomposition of audio-visual information

To test whether the condition modulation of speech MI could be attributed to a co-representation of visual lip information in the same ROI we calculated the conditional information between the MEG and the speech envelope, factoring out the encoding of temporal dynamics common to the speech and lip signals. With MI_speech&lip defined as MI(MEG;speech,lip), the CMI was defined as follows

$$CMI\ (MEG;speech|lip) = MI\_speech\&lip - MI\_lip \qquad (4)$$

where the first term on the right-hand side denotes the information carried by the local MEG signal about both the acoustic and visual speech, and the second term the MI about only the visual speech. The respective CMI values were then tested for significant condition effects (*Table 2*).

To further test whether the local representations of acoustic and visual speech in each ROI were independent or possibly strongly redundant (hence capturing the same aspect of sensory information), we computed a measure of normalized information theoretic redundancy during the VI condition as follows (*Belitski et al., 2010*; *Pola et al., 2003*; *et al., 2003*):

$$Red = (MI\_speech + MI\_lip - MI\_speech\&lip)/(MI\_speech + MI\_lip) * 100 \qquad (5)$$

This expresses redundancy as percentage of the total information that there would be in its absence of any redundancy. For these analysis both speech and lip signals were extracted at their respective optimal lag for each ROI/band and a common segment to each stimulus and the MEG activity was used for the calculation (segment duration = 60 s – 320 ms). Statistical tests contrasting condition-averaged information terms relied on the same RFX permutation framework and correction across all relevant dimensions as in all other analyses (FWE = 0.05). We compared condition-averaged MI_speech with MI_lip values using a two-sided test, contrasted condition-averaged redundancy values with their statistical bias (null-distribution), and tested for condition effects (GLM) on the CMI values.

## Analysis of condition effects on MEG signal amplitude

The amplitude within specific bands was defined as the absolute value of the instantaneous Hilbert-transformed band-pass MEG signal beamformed to each of the ROIs (c.f. *Figure 3E*). For each participant and experimental condition, we averaged the amplitude of the MEG time courses across time and repetitions of the same condition. Significance testing of condition changes in amplitude relied on the same RFX permutation-based approach as for the other modulation analyses, with maximum statistic multiple comparisons correction across ROIs and frequency bands (FWE = 0.05).

## Analysis of phase-amplitude couplings

We computed a measure of phase-amplitude coupling (PAC) between the oscillatory activity in different bands and regions. PAC was defined as

$$PAC = \sum_{t=1}^{N} A_{FH} * e^{i\theta_{FL}}/N \qquad (6)$$

where $A_{FH}$ and $\theta_{FL}$ denote the instantaneous Hilbert amplitude and phase angle of the high- and low-frequency MEG pass-band signal, respectively, and N is the number of time samples of the pass-band MEG signal in a specific condition. Low-frequency phase was extracted for the 0.25–1, 1–4, and 4–8 Hz bands. High-frequency amplitude was extracted for the 8–12, 12–18, 18–24, 24–36 and 30–48 Hz bands. We tested for both a significant condition-average PAC and for a significant modulation of PAC with conditions. Significance testing relied on the same RFX permutation-based approach as for the other modulation analyses, with maximum statistic correction for multiple comparisons across pairs of phase/power frequency pairs for the significance of condition averaged PAC, and also across pairs of phase/power ROIs for the GLM modulation (FWE = 0.05).

## Data sharing

The data analyzed for the ROI results presented in *Figures 2–5*, in the figure supplements and in *Tables 1–5*, as well as the speech and lip time courses analyzed in *Figure 1*, have been deposited on Dryad (doi:10.5061/dryad.j4567).

## Acknowledgements

We thank Hyojin Park for sharing audio-visual materials used to prepare the stimuli in this study.

## Additional information

### Funding

| Funder | Grant reference number | Author |
|---|---|---|
| Biotechnology and Biological Sciences Research Council | BB/L027534/1 | Christoph Kayser Joachim Gross |
| European Research Council | ERC-2014-CoG grant No 646657 | Christoph Kayser |
| Biotechnology and Biological Sciences Research Council | BB/M009742/1 | Bruno L Giordano Joachim Gross |
| Wellcome Trust | Joint Senior Investigator Grant No 098433 | Joachim Gross |
| Wellcome Trust | Senior Investigator Grant 107802/Z/15/Z | Philippe G Schyns |
| Wellcome Trust | Senior Investigator Award, UK; 107802 | Philippe G Schyns |
| Engineering and Physical Sciences Research Council | Multidisciplinary University Research Initiative USA/UK; 172046-01 | Philippe G Schyns |
| U.S. Department of Defense | Multidisciplinary University Research Initiative USA/UK; 172046-01 | Philippe G Schyns |
| Autonomous Province of Trento | Grandi Progetti 2012 Characterizing and Improving Brain Mechanisms of Attention-ATTEND | Stefano Panzeri |

The funders had no role in study design, data collection and interpretation, or the decision to submit the work for publication.

### Author contributions

BLG, Resources, Data curation, Software, Formal analysis, Validation, Investigation, Visualization, Methodology, Writing—original draft, Writing—review and editing; RAAI, Resources, Software, Methodology, Writing—review and editing; JG, Funding acquisition, Methodology, Writing—review and editing; PGS, Methodology, Writing—review and editing; SP, Conceptualization, Methodology, Writing—review and editing; CK, Conceptualization, Resources, Software, Formal analysis, Supervision, Funding acquisition, Validation, Investigation, Visualization, Methodology, Writing—original draft, Project administration, Writing—review and editing

### Author ORCIDs

Bruno L Giordano, http://orcid.org/0000-0001-7002-0486
Robin A A Ince, http://orcid.org/0000-0001-8427-0507
Stefano Panzeri, http://orcid.org/0000-0003-1700-8909
Christoph Kayser, http://orcid.org/0000-0001-7362-5704

## Ethics

Human subjects: The study was conducted in accordance with the Declaration of Helsinki and was approved by the local ethics committee (College of Science and Engineering, University of Glasgow). Written informed consent was obtained from all participants.

## Additional files

### Major datasets

The following dataset was generated:

| Author(s) | Year | Dataset title | Dataset URL | Database, license, and accessibility information |
|---|---|---|---|---|
| Giordano BL, Ince RAA, Gross J, Schyns PG, Panzeri S, Kayser C | 2017 | Data from: Contributions of local speech encoding and functional connectivity to audio-visual speech perception | http://dx.doi.org/10.5061/dryad.j4567 | Available at Dryad Digital Repository under a CC0 Public Domain Dedication |

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
