## [Decision Letter]

Thank you for submitting your article "Contributions of local speech encoding and functional connectivity to audio-visual speech integration" for consideration by *eLife*. Your article has been favorably evaluated by Richard Ivry (Senior Editor) and three reviewers, one of whom is a member of our Board of Reviewing Editors.. The reviewers have opted to remain anonymous.

The reviewers have discussed the reviews with one another and the Reviewing Editor has drafted this decision to help you prepare a revised submission.

Summary:

This study used MEG along with perceptual measures to address the question of how SNR and the informativeness of visual inputs combine to enhance both information encoding and the network interactions of brain areas processing speech cues. The authors highlight several results including 1) predictable feedforward auditory activation during high SNR, 2) visually mediated facilitation of auditory information representation in both auditory and premotor cortex, 3) an interaction of SNR and visual informativeness in in several frontal regions and 4) strong associated patterns of feedforward and feedback connectivity between auditory and premotor cortices. The behavioral benefits of viewing the speaker's face seemed best associated with the connectivity changes.

Essential revisions:

The reviewers identified a number of concerns and suggestions that the authors should explicitly address:

1) To disambiguate power and functional connectivity, and for other reasons (below), it would be very helpful if the authors would carefully detail the distribution of MEG power (by frequency) for each of the experimental conditions, but particularly for the increasing auditory SNR conditions. An additional comment below on this.

2) In general, the analysis lumps frequencies together a bit more than is ideal, the only division discussed at present is between the.25-1.0 Hz and the 1.0-4.0 Hz bands. Following from the power analysis asked for above, are there effects in other obvious bands (e.g., 4-7, etc.), and coupling interactions between bands? Do cross frequency interactions play a part in cross regional interactions?

3) The Discussion brings up the issue of predictive coding (citing the Arnal and Bastos studies). If the predictive coding account is to be invoked, it also makes sense to explore some of its predictions; e.g., the prediction error is the main component of the feedforward signal.

4) It would be helpful if the authors might be able to better link their findings to mechanisms of multimodal enhancement, e.g., phase reset vs. divisive normalization (van Atteveldt, 2015).

5) It's not clear what you mean by integration or enhancement. Normally, if you thought two different signals were being integrated, you would measure them each separately and then show that something non-linear happens when you combine them. MI is particularly well suited for this kind of analysis. You don't really show that here. For instance, it could be that in the low VI condition, subjects are simply not looking at the visual stimuli (there's no eye tracking as far as I can tell, so you can't say). This would make your comparison an AV vs. A comparison as opposed to a faulty integration interpretation. Furthermore, in the high VI case, it is unclear if two channels of sensory data are being integrated or if the brain is trading one channel for another. The easiest way forward I think, is to remove integration and keep enhancement.

6) It is concerning that you find such minimal (essentially non-existent) results in the LH. This is even true of the visual cortex! This may undermine the strength of the manuscript and make it a bit unclear what is actually happening. While you rightfully point out that your results may be expected given the frequency range of interest (<=4 Hz) and its relation to prosodic/syllabic information, I think this then requires a particular class of interpretation that I don't think is present in the current manuscript. Note that while you are correct to point out that various speech models (e.g. Hickok and Poeppel) suggest a rightward bias towards analyses on longer timescales, the models do indicate that this information is processed in both hemispheres, a result not demonstrated in the present manuscript. At the very least, I would reframe your interpretation to focus on what exactly you think the RH is doing here and why the LH is not doing it. If you think that audiovisual information is being integrated on longer timescales only, then say so. If you think that your measure only measures longer timescales and that this is why you only see RH effects, then say that.

7) The directed connectivity analysis is very interesting, but hard to interpret. It seems that in general, directed connectivity increases as a function of stimulus SNR (but again, this may be a by-product of neural SNR) and there is generally an effect of visual information, but not in all cases. The most interesting results are the interaction results (HGR->PMCL and SFGR->VCR) which show opposite effects. I think these results warrant more of an explanation especially given that they differ from the other results.

8) In your previous work (Park et al. 2016), you recognized that the visual information from AV speech is strongly correlated with the speech envelope. You rightfully incorporate this in your analysis for that manuscript, but not this one. One can make the argument that the purpose of that work was different (role of motor cortex vs. audio-visual integration), but I think then that you need to explain clearly what you have in mind when you say integration. What is exactly is being integrated and how? A clearer idea may also help the previous RH vs. LH issues listed above.

9) Related to this point it would also be helpful if this paper's findings were more explicitly separated from those of the authors' paper last year in *eLife* (Park et al).

10) It would be helpful if a supplementary analysis were done for each of the behavioral results that uses d' instead of probability of correct answers.

11) Although the main conclusions about the SNR effects are convincing, the conclusions related to the multisensory effects are not. It is not ruled out that the multisensory effects could be purely caused by lip reading effects. Recently, a series of papers by Edmund Lalor's group showed that a talking face alone can lead to neural entrainment to the (not presented) speech envelope. Previous work by Luo et al. showed a similar effect.

---

## [Author Response]

*Essential revisions:*

*The reviewers identified a number of concerns and suggestions that the authors should explicitly address:*

*1) To disambiguate power and functional connectivity, and for other reasons (below), it would be very helpful if the authors would carefully detail the distribution of MEG power (by frequency) for each of the experimental conditions, but particularly for the increasing auditory SNR conditions. An additional comment below on this.*

To address this point we have systematically analyzed the power of oscillatory activity and potential changes of this across the experimental conditions. We implemented this analysis for each of the frequency bands investigated in the speech entrainment analysis and focused on the ROIs used in the functional connectivity analysis, i.e., those voxels that exhibited interesting condition effects in speech entrainment. This analysis is detailed in the Methods (section ‘Analysis of condition effects on MEG amplitude’). The respective results are presented in a separate section in the Results (section ‘Changes in speech entrainment are not a result of changes in oscillatory amplitude’), Table 5and Figure 3—figure supplement 3. They reveal a suppression of power during the VI condition in several ROIs and bands, but no significant SNR or SNRxVIVN effects on power. Importantly, none of these changes in the strength of oscillatory activity overlaps spectrally and anatomically with the condition effects on speech entrainment reported in our main results (Figure 3); and only the change in functional connectivity between pSTG and IFG (Figure 4) overlaps with a change in pSTG‐R power in the 4‐8 Hz range. All in all, these results suggest that the reported visual enhancement of speech encoding and connectivity are unlikely to be systematically linked to changes in the strength of oscillatory activity, as we now conclude in the respective section of the Results.

*2) In general, the analysis lumps frequencies together a bit more than is ideal, the only division discussed at present is between the.25-1.0 Hz and the 1.0-4.0 Hz bands. Following from the power analysis asked for above, are there effects in other obvious bands (e.g., 4-7, etc.), and coupling interactions between bands? Do cross frequency interactions play a part in cross regional interactions?*

The focus on the lowest frequency bands (0.25‐1 and 1‐4Hz) in the Results and Discussion is a genuine finding, not a selection bias in our approach. As outlined in the Methods, we followed a data driven approach and quantified speech entrainment and condition effects across a wide range of frequency bands, from 0.25Hz to 48Hz. The width of these bands increased with frequency, as is common practice in filtering and wavelet approaches. By Heisenberg’s theorem, using more narrow bands will necessarily compromise temporal resolution, which however is important to quantify reliably the temporal similarity between speech and brain signals (Gross JNeurosciMethods’ 14). Here the specific choice of bands and bandwidths was driven by our experience in analyzing band‐limited field potentials and neuroimaging signals (Kayser et al. Neuron ‘09, Belitski JCompNeurosci’ 10; Ng et al. CerCor’ 12; Keitel et al. Neuroimage ‘16), which maximizes sensitivity without compromising frequency resolution, and ensures computational tractability, as closer frequencies tend to be highly correlated and increase computational burden without providing any additional gain in knowledge.

To address the second part of the question we have implemented a systematic analysis of cross‐frequency interactions as characterized by phase‐amplitude coupling (PAC). We focused on PAC based on the phase in low frequencies (up to 8Hz) and the power in higher bands (above 8 Hz). The relevant details are presented in the Methods (section ‘Analysis of phase‐amplitude coupling’) and the findings are presented in the Results (section ‘Changes in directed connectivity do not reflect changes in phase‐amplitude coupling’) and Table 6. While we observed significant PAC across a number of bands and ROIs, we did not find significant changes in PAC with the experimental conditions, suggesting that the condition‐ changes in functional connectivity are not systematically related to specific patterns of cross‐ frequency coupling, as we now conclude in the Results.

*3) The Discussion brings up the issue of predictive coding (citing the Arnal and Bastos studies). If the predictive coding account is to be invoked, it also makes sense to explore some of its predictions; e.g., the prediction error is the main component of the feedforward signal.*

We agree that the mentioning of predictive coding in the previous submission was rather unspecific and hypothetical. Unfortunately the current experimental design does not allow a direct test of such models and their predictions, as no experimentally quantifiable prediction error or manipulation of such was included. However, given the prevailing notion that lip reading holds predictive signals for acoustic speech (Chandrasekaran PlosComputBiol’ 09; Bernstein SpeechComm ‘04), we expanded the Discussion to provide a more in depth reflection of this issue, as well as to better link the present results to previous work on predictive coding along the auditory pathways (extended first paragraph in the section ‘The mechanistic underpinnings of audio‐visual speech encoding’ of the Discussion).

*4) It would be helpful if the authors might be able to better link their findings to mechanisms of multimodal enhancement, e.g., phase reset vs. divisive normalization (van Atteveldt, 2015).*

We have added a section discussing potential mechanisms to the Discussion. However, similar to previous studies (e.g. van Attefeldt Neuron 2014) we have to remain vague, as our data don’t directly speak on the underlying neural mechanisms (last paragraph of the Discussion section ‘The mechanistic underpinnings of audio‐visual speech encoding’).

*5) It's not clear what you mean by integration or enhancement. Normally, if you thought two different signals were being integrated, you would measure them each separately and then show that something non-linear happens when you combine them. MI is particularly well suited for this kind of analysis. You don't really show that here. For instance, it could be that in the low VI condition, subjects are simply not looking at the visual stimuli (there's no eye tracking as far as I can tell, so you can't say). This would make your comparison an AV vs. A comparison as opposed to a faulty integration interpretation. Furthermore, in the high VI case, it is unclear if two channels of sensory data are being integrated or if the brain is trading one channel for another. The easiest way forward I think, is to remove integration and keep enhancement.*

We agree that it is important to define these terms well, in light of the sometimes discrepant use of the term integration (Stein EuroJNeurosci ‘10). First, in the revised manuscript we now avoid the term integration, as our study does not provide direct evidence for the integration of the same speech feature across auditory and visual domains, as would be required to make a direct claim pertaining to multisensory feature integration. As a consequence, we have also modified the title of this submission.

Second, to enhance the impact of this study along these lines and to address this and other comments (points 8, 11 below), we have added several additional analyses. We have analyzed the encoding of lip movement signals and we have used conditional mutual information and information theoretic redundancy to quantify speech encoding while discounting influences of lip movements. These newly added results are presented in new sections of the Results (‘Noise invariant dynamic representations of lip movements’ and ‘Speech entrainment does not reflect trivial entrainment to lip signals’), Table 2, and a new Figure 3—figure supplement 2. They demonstrate that the entrainment of brain activity to the acoustic speech envelope is largely reflecting a visually‐informed but genuine acoustic representation. Our findings hence speak in favor of an interpretation in which the reported local effects reflect changes within an acoustically driven speech representation that is influenced by visual and acoustic context, rather than a bimodal audio‐visual representation of acoustic and visual speech information within the respective ROIs. Yet, given the lack of clear evidence for the integration of the same feature or linguistic information within each ROI in the present data, we refrain from interpreting our results in the light of multisensory integration and hence avoid this term.

We have also revised the Discussion to more clearly interpret our results in the context of the newly added analyses (in section ‘Entrained auditory and visual speech representations in temporal, parietal and frontal lobes’).

*6) It is concerning that you find such minimal (essentially non-existent) results in the LH. This is even true of the visual cortex! This may undermine the strength of the manuscript and make it a bit unclear what is actually happening. While you rightfully point out that your results may be expected given the frequency range of interest (<=4 Hz) and its relation to prosodic/syllabic information, I think this then requires a particular class of interpretation that I don't think is present in the current manuscript. Note that while you are correct to point out that various speech models (e.g. Hickok and Poeppel) suggest a rightward bias towards analyses on longer timescales, the models do indicate that this information is processed in both hemispheres, a result not demonstrated in the present manuscript. At the very least, I would reframe your interpretation to focus on what exactly you think the RH is doing here and why the LH is not doing it. If you think that audiovisual information is being integrated on longer timescales only, then say so. If you think that your measure only measures longer timescales and that this is why you only see RH effects, then say that.*

The comment identifies an important issue that still poses controversies in the current literature. The data presented in the previous submission exhibited significantly stronger speech entrainment in the right vs. the left hemisphere (revised Figure 3—figure supplement 1), and we had reported significant condition effects mostly within the right hemisphere. However, this had left it unclear whether there was indeed a significant difference between hemispheres in terms of the strength of the condition effects (as expressed by the GLM betas). We have performed an additional analysis to directly test this, which is now presented in the Results (section ‘Condition effects are hemisphere‐dominant but not strictly lateralized’). This revealed that only one of the reported speech MI effects in Figure 3 is significantly lateralized (the SNR effect in IFGop‐R); presented in new Table 2. This suggests that care should be taken when interpreting the dominance of significant condition effects in the RH as evidence for a significant lateralization.

Furthermore, we would like to note that the apparent lateralization of speech related activity as reported in the literature potentially depends on a large number of factors, incl. the nature of the recorded brain signal (fMRI, MEG frequency bands), the type of the stimulus material (single words, full sentences), and the specific statistical contrast used (activation vs. rest, or a specific experimental effect such as the encoding of word identity). For example, several studies have specifically implied the right hemisphere in the encoding of connected speech (Alexandrou Neuroimage’ 17; Horowitz‐Kraus ‘15), or demonstrated a stronger involvement of low‐frequency activity or low‐frequency speech‐entrainment in the right hemisphere (Giraud Neuron’ 07; Gross et al. PlosBiol’ 13). Given the previous literature it is not surprising to us to observe a dominance of the right hemisphere in the present study, which uses continuous speech.

Finally, we agree with the reviewers that the observed dominance of the RH could possibly result from the use of speech‐to‐brain entrainment as an index of speech encoding in the present study. It is possible that this measure naturally emphasizes longer time scales (e.g. below 12Hz) given their dominance on signal power (c.f. Figure 1, speech envelope power plot). Given the possibility that the RH is particularly involved in processing speech information on these time scales, the methodological approach used here may contribute to the observed RH dominance.

We revised the Discussion to include these considerations and the new results (new section ‘A lack of evidence for lateralized representations’).

7) The directed connectivity analysis is very interesting, but hard to interpret. It seems that in general, directed connectivity increases as a function of stimulus SNR (but again, this may be a by-product of neural SNR) and there is generally an effect of visual information, but not in all cases. The most interesting results are the interaction results (HGR->PMCL and SFGR->VCR) which show opposite effects. I think these results warrant more of an explanation especially given that they differ from the other results.

In the Discussion we emphasize the positive main effects of SNR and VIVN, in particular as these relate to the behavioral data and hence are the most important and straightforward to interpret. However, we do agree that these mentioned (negative) interactions are interesting and warrant explicit consideration. We have extended the Discussion to elaborate on these in more detail.

*8) In your previous work (Park et al. 2016), you recognized that the visual information from AV speech is strongly correlated with the speech envelope. You rightfully incorporate this in your analysis for that manuscript, but not this one. One can make the argument that the purpose of that work was different (role of motor cortex vs. audio-visual integration), but I think then that you need to explain clearly what you have in mind when you say integration. What is exactly is being integrated and how? A clearer idea may also help the previous RH vs. LH issues listed above.*

This is an important point, which we have addressed by substantial additional data analysis (as also mentioned under point 5 above). We have revised the Methods to provide all the relevant details (section ‘Decomposition of audio‐visual information’).

First, we characterized the temporal dynamics of lip movements, as in our previous study (Park et al., 2016), and systematically quantified the lip‐to‐brain entrainment in the same way as the speech‐to‐brain entrainment. The Results (section ‘Noise invariant dynamic representations of lip movements’) and Figure 3—figure supplement 1, Figure 3—figure supplement 2, and Table 2, reveal the wide‐spread entrainment to dynamic lip signals in visual cortex and the temporal lobe. However, we did not find significant changes in the lip encoding across SNRs. Furthermore, we performed a direct comparison of the relative strength of acoustic and lip MI within the regions of interest (Table 2). This revealed stronger encoding of acoustic speech across many ROIs and bands, but a dominance of lip representations in visual cortex. Consequently, the revised manuscript now provides a comparative analysis of neural entrainment to acoustic (speech envelope) and visual (lip movements) speech signals at the same time scales, and how each of these is affected by acoustic SNR.

Second, we now include two analyses of the information theoretic dependency between acoustic and visual speech representations (section ‘Speech entrainment does not reflect trivial entrainment to lip dynamics). The results (Figure 3—figure supplement 2, Table 2)demonstrate that the entrainment of brain activity to the acoustic speech envelope within the considered ROIs is not trivially explained by an overlap of separate representations of the acoustic and visual speech signals, but rather reflects a visually informed, but genuinely acoustic representation. Importantly, all of the condition effects on speech entrainment reported in Figure 3 persisted when factoring out direct influences of lip movements using conditional mutual information (Table 2). Furthermore, the redundancy between representations of acoustic and visual speech was small, and only significant in association cortices.

These results have provided several important new insights: First, early sensory regions seem to contain largely genuine representations of acoustic speech, while association regions contain overlapping representations of acoustic speech and lip movements. Second, the representation of acoustic speech in visual cortex is not a trivial consequence of the local representation of lip movements information (which we describe by the additional analysis), but rather seems to reflect an independent representation of acoustically informed speech. These insights have been included in the revised Discussion (subsection “Multisensory enhancement of speech encoding in the frontal lobe”).

*9) Related to this point it would also be helpful if this paper's findings were more explicitly separated from those of the authors' paper last year in eLife (Park et al).*

We now better differentiate the current and this previous study, which has become even more important given that the revised manuscript in part replicates some of the findings reported in Park et al. 2016 (subsections “Entrained auditory and visual speech representations in temporal, parietal and frontal lobes” and “Multisensory behavioral benefits arise from distributed network mechanisms”).

*10) It would be helpful if a supplementary analysis were done for each of the behavioral results that uses d' instead of probability of correct answers.*

By design of our experimental paradigm the specific words that were part of the presented story (the signal in detection theory) were associated with specific experimental conditions; however, the words that were not part of the story (the noise) were not. As a consequence, whereas hit rate varied across conditions, false alarm rate was bound to be constant across conditions. Under these circumstances, condition‐specific d’ is strongly correlated with condition‐specific performance. Specifically, the group‐average correlation between condition‐specific performance and d’ was 0.97 (SEM = 0.06) and highly significant (T(18) = 32.57, p < 10^-6^). As a result, it would have made little difference to compute the neuro‐behavioral correlations with d’ rather than the probability correct. We have added this information to the Methods (section ‘Experimental design and stimulus presentation’).

*11) Although the main conclusions about the SNR effects are convincing, the conclusions related to the multisensory effects are not. It is not ruled out that the multisensory effects could be purely caused by lip reading effects. Recently, a series of papers by Edmund Lalor's group showed that a talking face alone can lead to neural entrainment to the (not presented) speech envelope. Previous work by Luo et al. showed a similar effect.*

This relates to point 8 above, which we have addressed with substantial additional analysis as outlined there.